# Cyclin-dependent kinase 5 (Cdk5) activity is modulated by light and gates rapid phase shifts of the circadian clock

**Andrea Brenna[1,2]\*, Micaela Borsa[3,4], Gabriella Saro[1], Jürgen A Ripperger[1], Dominique A Glauser[1], Zhihong Yang[2], Antoine Adamantidis[3,4], Urs Albrecht[1]\***

[1]Department of Biology, University of Fribourg, Fribourg, Switzerland; [2]Department of Endocrinology, Metabolism, and Cardiovascular System, Section of Medicine, University of Fribourg, Fribourg, Switzerland; [3]Zentrum für Experimentelle Neurologie, Department of Neurology, Inselspital, Bern University Hospital, University of Bern, Bern, Switzerland; [4]Department of Biomedical Research, University of Bern, Bern, Switzerland

**\*For correspondence:**
andrea.brenna@unifr.ch (AB);
urs.albrecht@unifr.ch (UA)

**Competing interest:** The authors declare that no competing interests exist.

## eLife Assessment

This **important** chronobiological study in mice suggests that light modulated activity of Cdk5 activity on the PKA-CaMK-CREB signaling pathway provides missing molecular mechanistic details to understand light-induced circadian clock phase delays during the early night, but not for phase advances in the morning. The authors provide **convincing** evidence bridging from behavioral to molecular/cellular experiments to neural activity imaging.

**Abstract** The circadian clock enables organisms to synchronize biochemical and physiological processes over a 24 hr period. Natural changes in lighting conditions, as well as artificial disruptions like jet lag or shift work, can advance or delay the clock phase to align physiology with the environment. Within the suprachiasmatic nucleus (SCN) of the hypothalamus, circadian timekeeping and resetting rely on both membrane depolarization and intracellular second-messenger signaling. Voltage-gated calcium channels (VGCCs) facilitate calcium influx in both processes, activating intracellular signaling pathways that trigger *Period* (*Per*) gene expression. However, the precise mechanism by which these processes are concertedly gated remains unknown. Our study in mice demonstrates that cyclin-dependent kinase 5 (Cdk5) activity is modulated by light and regulates phase shifts of the circadian clock. We observed that knocking down Cdk5 in the SCN of mice affects phase delays but not phase advances. This is linked to uncontrolled calcium influx into SCN neurons and an unregulated protein kinase A (PKA)-calcium/calmodulin-dependent kinase (CaMK)-cAMP response element-binding protein (CREB) signaling pathway. Consequently, genes such as *Per1* are not induced by light in the SCN of Cdk5 knock-down mice. Our experiments identified Cdk5 as a crucial light-modulated kinase that influences rapid clock phase adaptation. This finding elucidates how light responsiveness and clock phase coordination adapt activity onset to seasonal changes, jet lag, and shift work.

## Introduction

The circadian system coordinates biochemical and physiological functions in our body, synchronizing them with the environmental day-night cycle. Misalignment of the internal body clock with the external light-dark (LD) cycle, caused by shift work or jet lag, leads to inefficient regulation of body functions.

**eLife digest** Our bodies evolved to follow daily rhythms, influencing our sleeping and waking patterns and our usual mealtimes. These rhythms are based on circadian clocks that allow our bodies to stay in tune with the day-night cycle in our environment.

Circadian rhythms are controlled by a set of biological mechanisms termed molecular clocks, which are found in every cell and organ. The body also has a 'master clock', which is in a part of the brain called the suprachiasmatic nuclei (SCN). The SCN is the body's main 'timekeeper' and coordinates all our daily cycles of biological processes and behaviours.

Molecular clocks also respond to artificial stimuli, which can cause shifts in circadian rhythms. These alterations disrupt the normal alignment of our bodily processes with the environmental day-night cycle, meaning that our bodies work less efficiently. For example, light exposure early during the night changes our circadian rhythms so that we go to sleep and wake up later, a phenomenon called phase delay.

The enzyme CDK5 is part of the SCN's master clock. CDK5 helps control normal circadian rhythms: it is more active in the dark (i.e., at night), when it helps to turn off genes that respond to light, and is inactive in light conditions, ensuring that the light response genes stay switched on during the day. Since CDK5 is also involved in many neurological diseases linked to disturbed circadian rhythms, Brenna et al. wanted to determine whether it also controlled the circadian shift caused by light exposure early during the night.

To simulate this mistimed light, mice were exposed to 15 minutes of bright light two hours after the onset of darkness in the laboratory light/dark cycle. Biochemical and genetic analysis revealed that in standard mice, this reduced CDK5 activity in the SCN, switched light response genes back on, and resulted in phase delay. However, light exposure did not cause any shift in behaviour in genetically engineered mice lacking CDK5, confirming that CDK5 was indeed responsible for the phase delay observed.

These results contribute to our understanding of the mechanisms behind the body's response to stimuli that force our internal clock out of sync with our environment. Brenna et al. hope that targeting CDK5 may one day help us cope better with circadian misalignment and the health problems associated with it, especially for people affected by jet lag or shift work.

---

Consequently, circadian misalignment can result in obesity, cancer, addictive behaviors, cardiovascular disease, and neurological disorders (*Roenneberg and Merrow, 2016*). Therefore, it is crucial to understand how the environment impacts the clock and how these entities interact.

In mammals, the master circadian clock is located in the ventral part of the hypothalamus, just above the optic chiasm, in the suprachiasmatic nuclei (SCN). These nuclei coordinate daily cycles of physiology and behavior (*Hastings et al., 2019*). Molecular daily oscillations are generated at the cellular level by a cell-autonomous transcription-translation feedback loop (TTFL) involving a set of clock genes (*Takahashi, 2017*) and post-translational modifiers such as kinases (*Hirano et al., 2016*; *Partch, 2020*). Circuit-level interactions among SCN cells produce a coherent daily oscillation (*Allen et al., 2017*), which can be modulated by light signals to match the environmental LD cycle. Light is perceived by melanopsin-containing intrinsically photosensitive retinal ganglion cells in the eye, and the signal produced in these cells travels via the retinohypothalamic tract (RHT) to the SCN (*LeGates et al., 2014*). The release of glutamate at the RHT terminals stimulates AMPA/NMDA receptors, leading to $Ca^{2+}$ influx into the SCN (*Ding et al., 1994*). Additionally, the activity of various kinases is altered, including DARPP-32 (dopamine and cAMP-regulated phosphoprotein of 32 kD), PKA (protein kinase A), and CaMK ($Ca^{2+}$/calmodulin-dependent kinases). This cascade culminates in the phosphorylation of CREB (cyclic AMP [cAMP] response element-binding protein) (*Obrietan et al., 1998*; *Ginty et al., 1993*; *Gau et al., 2002*; *Yan et al., 2006*; *Wheaton et al., 2018*). This event promotes chromatin phosphorylation (*Crosio et al., 2000*) and acetylation (*Brenna et al., 2021*) via the recruitment of CRTC1 (cAMP-regulated transcriptional co-activator 1) and the histone acetyltransferase CBP (CREB-binding protein), involving the clock protein PER2. Consequently, immediate-early gene and clock gene expression is induced (*Rusak et al., 1990*; *Albrecht et al., 1997*; *Shigeyoshi et al., 1997*), causing a phase shift of the TTFL in oscillating cells of the SCN (*Allen et al., 2017*). This manifests at

the behavioral level as a change of locomotor activity onset (phase shift) the day after the light pulse. The direction of the phase shift depends on the clock's temporal state. Light perceived in the early night promotes phase delays, while a light pulse late at night promotes phase advances. Light in the middle of the day does not alter the clock phase (*Daan and Pittendrigh, 1976*). For this so-called resetting of the circadian clock, the *Per1* and *Per2* genes appear to be important in mice. While *Per1* is essential for phase advances, *Per2* function is necessary for phase delays (*Albrecht et al., 2001*; *Spoelstra et al., 2004*).

Voltage-gated calcium channels (VGCCs) are classified into high voltage-activated channels, which include L-type, and low voltage-activated subtypes, also known as T-type channels (*Catterall, 2011*). T-type VGCCs are involved in phase delays, whereas L-type VGCCs are related to phase advances (*Kim et al., 2005*; *Schmutz et al., 2014*). $Ca_V3.1$, $Ca_V3.2$, and $Ca_V3.3$ belong to the T-type channel family, which is critically important for neuronal excitability (*Perez-Reyes, 2003*). The activity of these T-type channels is regulated by various kinases, including PKA (*Kim et al., 2006*), PKC (*Chemin et al., 2007*), and Cdk5 (cyclin-dependent kinase 5) (*Calderón-Rivera et al., 2015*).

Cdk5 is a proline-directed serine/threonine kinase that forms a complex with its neural activators p35 or p39 (*Tang et al., 1995*; *Tsai et al., 1994*) and cyclin I (*Brinkkoetter et al., 2009*). The complex of Cdk5 and its activators controls various neuronal processes such as neurogenesis, neuronal migration, and synaptogenesis (*Jessberger et al., 2009*; *Kawauchi, 2014*). In vivo and in vitro experiments show that Cdk5 kinase activity is low in the light phase and high in the dark phase (*Brenna et al., 2019*; *Ripperger et al., 2022*). It regulates the circadian clock in the SCN via phosphorylation of PER2 at serine 394. Upon phosphorylation by Cdk5, PER2 is stabilized and enters the nucleus to participate in the regulation of the TTFL and CREB-related transcriptional events (*Brenna et al., 2021*; *Brenna et al., 2019*). Since Cdk5 regulates the T-type channel $Ca_V3.1$ (*Calderón-Rivera et al., 2015*) and the circadian clock via PER2 phosphorylation (*Brenna et al., 2019*), we analyzed the potential role of Cdk5 in the light-mediated clock resetting mechanism.

## Results

### *Cdk5* knock-down in the SCN impairs light-induced phase delays

Light perceived during the dark period elicits changes in the clock phase (*Daan and Pittendrigh, 1976*). To test whether *Cdk5* plays a role in this process, we knocked down *Cdk5* in the SCN via stereotaxic application of adeno-associated viruses (AAVs). We injected an adenovirus expressing shRNA to silence *Cdk5* (sh*Cdk5*) and, as a control, an adenovirus expressing a control shRNA (scr) into the SCN (*Brenna et al., 2019*). Consistent with our previous observations (*Brenna et al., 2019*), we found that silencing *Cdk5* in the SCN reduced its expression in the SCN (*Figure 1—figure supplement 1a*) and the expression of PER2 (*Figure 1—figure supplement 1b*). Under constant darkness (DD) conditions, this knock-down of *Cdk5* shortened the clock period in male mice, as assessed by wheel-running activity (*Figure 1a and b*, *Figure 1—figure supplement 1c*). This period was not influenced by light pulses (*Figure 1—figure supplement 1d*). However, the onset of activity was affected after releasing mice into constant darkness (DD). Light at zeitgeber time (ZT) 14 (where ZT0 is lights on and ZT12 is lights off) delayed the clock phase, whereas light at ZT22 advanced it in control (scr) animals, with light at ZT10 having no effect (*Figure 1a and c*, Aschoff type II protocol). The animals with silenced *Cdk5* in the SCN (sh*Cdk5*) behaved similarly to controls (scr), except for ZT14. Light did not elicit a phase delay at this time, suggesting that *Cdk5* plays a role in the phase delay mechanism. Similar results were obtained for female animals (*Figure 1—figure supplement 1e–g*).

To corroborate our observations, we performed the same experiment in DD (Aschoff type I protocol). The sh*Cdk5* animals displayed a shorter period compared to scr controls (*Figure 1d and e*), consistent with previous observations (*Brenna et al., 2019*). After determining each animal's clock period, we administered light pulses of 15 min at circadian times (CT) 10, CT14, and CT22 for each animal (orange stars, *Figure 1d*). Light at CT10 had no effect on both the sh*Cdk5* and scr control mice (*Figure 1f*). Light applied at CT14 promoted a phase delay in scr control mice. However, silencing of *Cdk5* impaired the delay of the clock phase (*Figure 1d and f*), which is consistent with the observation at ZT14 (*Figure 1a and c*). Light at CT22 elicited normal phase advances in sh*Cdk5* and scr controls (*Figure 1d and f*), similar to the light pulse applied at ZT22 (*Figure 1a and c*). From these

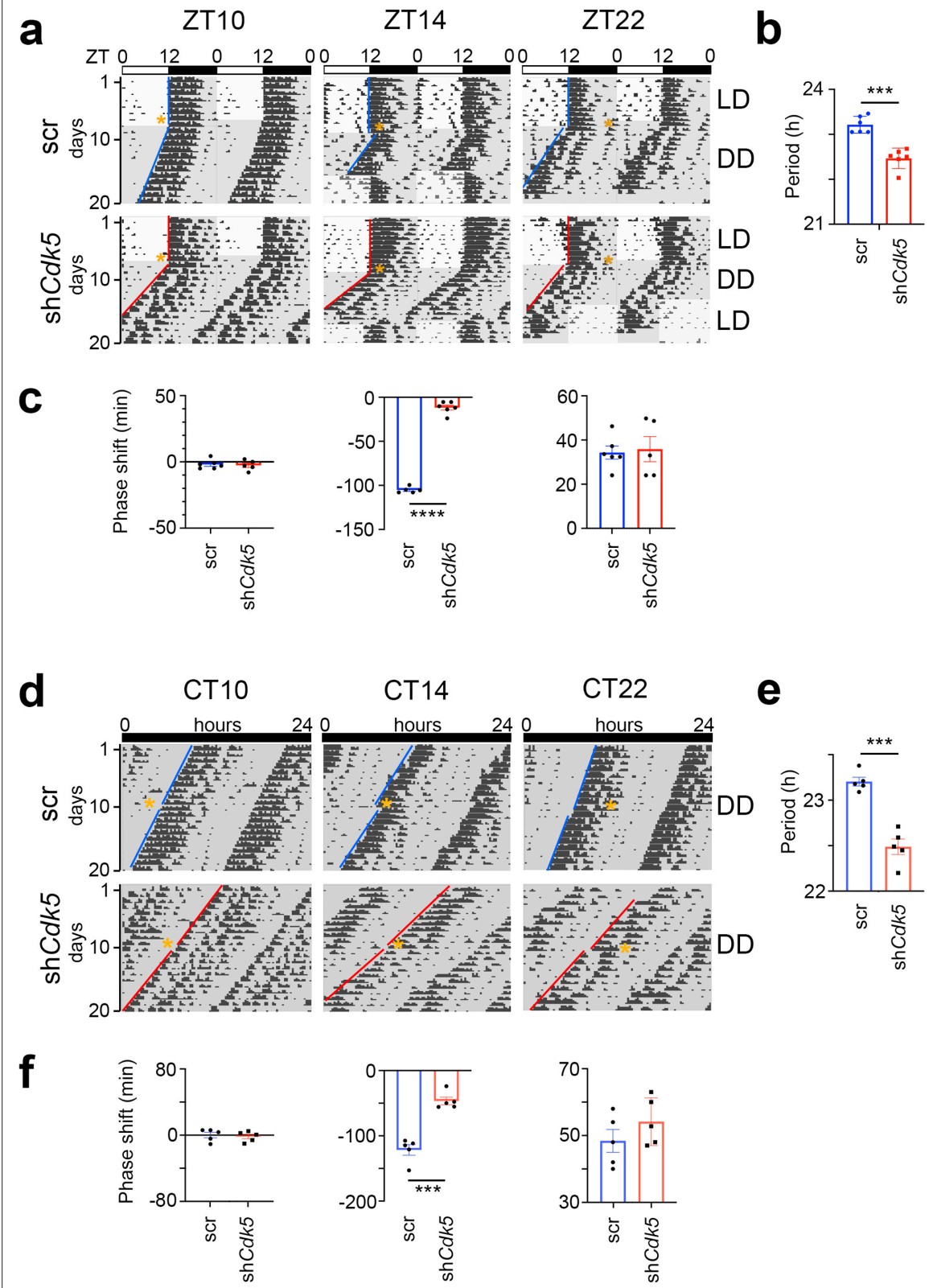

**Figure 1.** Knock-down of Cdk5 in the suprachiasmatic nucleus (SCN) shortens period and reduces phase delays but not phase advances. (**a**) Examples of double-plotted wheel-running actograms of control (scr) and *Cdk5* knock-down (sh*Cdk5*) male mice. Animals were kept under a 12 hr light/12 hr dark cycle (white and gray areas, respectively) (LD). After 8–10 days they received a 15 min light pulse at the indicated zeitgeber times (ZT) (yellow stars). After the light pulse animals were released into constant darkness (DD). This light pulse assessment is termed Aschoff type II. (**b**) Circadian period (τ) of

*Figure 1 continued on next page*

*Figure 1 continued*

sh*Cdk5* mice (red) is significantly shorter compared to scr controls (blue). τ scr = 23.21 ± 0.08 hr, τ sh*Cdk5*=22.47 ± 0.09 hr. All values are mean ± SEM, unpaired t-test with Welch's correction, n=6, ***p<0.001. (**c**) Quantification of phase shifts (φ) after a 15 min light pulse at ZT10, ZT14, and ZT22. The phase shift at ZT14 is strongly reduced in sh*Cdk5* animals (red) compared to scr controls (blue). scr: φ ZT10: –1.93±1.43 min, φ ZT14: –105.24±1.54 min, φ ZT22: 34.30±2.97 min, sh*Cdk5*: φ ZT10: –2.60±1.72 min, φ ZT14: –11.80±2.81 min, φ ZT22: 35.88±5.68 min. All values are mean ± SEM, unpaired t-test with Welch's correction, n=5–6, ****p<0.0001. (**d**) Examples of double-plotted wheel-running actograms of control (scr) and *Cdk5* knock-down (sh*Cdk5*) mice. Animals were kept under DD. After 8–10 days they received a 15 min light pulse at the indicated circadian times (CT) (orange stars). This light pulse assessment is termed Aschoff type I. (**e**) Circadian period of sh*Cdk5* mice (red) is significantly shorter compared to scr controls (blue). τ scr = 23.20±0.05 hr, τ sh*Cdk5*=22.48±0.09 hr. All values are mean ± SEM, unpaired t-test with Welch's correction, n=5, ***p<0.001. (**f**) Quantification of phase shifts after a 15 min light pulse at CT10, CT14, and CT22. The phase shift at CT14 is strongly reduced in sh*Cdk5* animals (red) compared to scr controls (blue). scr: φ ZT10: 0.12±3.31 min, φ ZT14: –121.52±8.18 min, φ ZT22: 48.40±3.43 min, sh*Cdk5*: φ ZT10: –1.68±2.78 min, φ ZT14: –46.60±5.84 min, φ ZT22: 54.16±3.19 min. All values are mean ± SEM, unpaired t-test with Welch's correction, n=5, ***p<0.001.

The online version of this article includes the following figure supplement(s) for figure 1:

**Figure supplement 1.** Validation of knock-down of Cdk5 expression.

experiments, we conclude that *Cdk5* plays a role in delaying the clock phase in response to a light pulse in the early activity period of mice.

## Cdk5 activity is modulated by light in the early night

Given that Cdk5 plays a significant role in the phase shift of the circadian clock, we investigated whether the light signal at ZT14 could affect the levels of Cdk5 and its co-activator p35 in the SCN. To this end, we collected SCN samples at ZT14 in the dark or after a 15 min light pulse and performed a western blot on total protein extracts. To ensure proper light induction, we measured the light-dependent phosphorylation of PKA (*Figure 2a and b*), CaMKII and CREB (*Figure 2—figure supplement 1a–d*). We confirmed that PKA, CaMKII, and CREB phosphorylation levels increased in response to light in the SCN (*Figure 2a and b*, *Figure 2—figure supplement 1a–d*). Interestingly, we observed that light could also increase the p35 protein level, although the levels of Cdk5 remained unaffected (*Figure 2a and c*). Given the increase in p35 levels due to light, we wondered whether this event would affect the kinase activity of the Cdk5/p35 complex. We performed an in vitro kinase assay using immunoprecipitated Cdk5 from SCN tissue collected from mice either not exposed to light or exposed to light at ZT14. We used the recombinant histone H1 as a substrate in the presence of radioactive ATP (*Brenna et al., 2019*). Surprisingly, our results indicated that Cdk5 kinase activity decreased in response to light (*Figure 2d and e*), suggesting that light may affect the interaction between Cdk5 and p35. To test this hypothesis, we performed a co-immunoprecipitation experiment using an antibody against Cdk5. Our results revealed that SCN extracts from mice that received a light pulse at ZT14 contained less p35 in a complex with Cdk5 (*Figure 2f*).

Taken together, the results support the hypothesis that light affects Cdk5 activity by interfering with the formation of the Cdk5/p35 complex. Interestingly, the light pulse at ZT14 might affect more than just Cdk5/p35 protein-protein interactions, potentially involving additional unknown proteins (*Figure 2—figure supplement 1e*).

## Cdk5 impacts the CREB signaling pathway via CaMK

Deletion of a cAMP-responsive element (CRE) in the *Per1* promoter blunted light-induced *Per1* expression in the SCN at night (*Ikegami et al., 2020*). Because nocturnal light induces phosphorylation of CREB and phosphorylated CREB (p-CREB) can bind to CREs (*Naruse et al., 2004*; *Travnickova-Bendova et al., 2002*; *Tischkau et al., 2003*), we investigated whether Cdk5 participates in the pathway evoking the CREB phosphorylation at serine-133 (pSer-133), a site known to be involved in phase delays, and *Per1* induction (*Gau et al., 2002*). Therefore, we performed immunohistochemical analysis using an antibody detecting phosphate on CREB at serine 133 (p-CREB-S133) (*Figure 3a*, *Figure 3—figure supplement 1b*, control *Figure 3—figure supplement 1c*). In the SCN of control animals (scr), we observed p-CREB-S133 in nuclei of neurons after the light was delivered at ZT14 but not in controls (*Figure 3a*, arrowheads). In contrast, p-CREB-S133 was already detected in nuclei before the light pulse in sh*Cdk5* animals (*Figure 3a*, arrowheads), indicating that *Cdk5* plays a role in gating the phosphorylation of CREB.

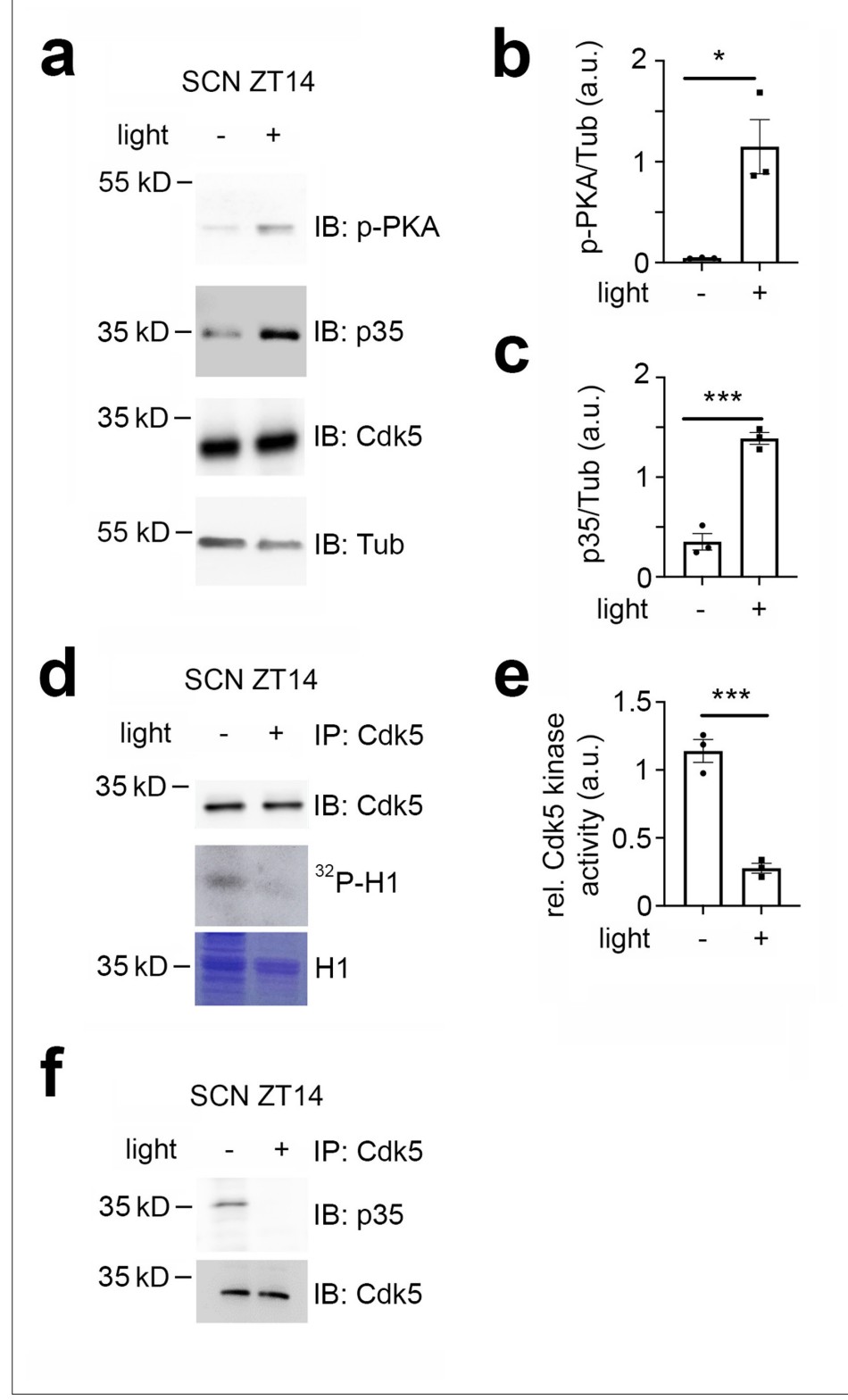

**Figure 2.** Cdk5 activity is modulated by light in the early night. Immuno-western blotting (IB), immunoprecipitation (IP), and Cdk5 kinase activity assays from suprachiasmatic nucleus (SCN) tissue extracts harvested 30 min after light (+) and no light (-) given at zeitgeber time (ZT) 14. (**a**) Western blot depicting the amounts of phospho PKA (p-PKA), p35 co-activator, Cdk5, and tubulin (Tub, control) before and after light pulse at ZT14. (**b**) Quantification of p-PKA relative to tubulin. Values are the mean ± SEM. Unpaired t-test, n=3, *p<0.05. (**c**) Quantification of p35

*Figure 2 continued on next page*

*Figure 2 continued*

co-activator of Cdk5. Values are the mean ± SEM. Unpaired t-test, n=3, ***p<0.001. (**d**) Cdk5 kinase activity assay. IP of SCN extracts with antibodies against Cdk5 showing the presence of Cdk5 (upper panel) and total protein with Coomassie blue staining (lower panel) as a control for the presence of H1. The middle panel depicts histone H1 phosphorylated by Cdk5, visualized as $^{32}$P-histone H1 ($^{32}$P-H1). (**e**) Quantification of Cdk5 kinase activity relative to H1 levels. Values are the mean ± SEM. Unpaired t-test, n=3, ***p<0.001. (**f**) Co-immunoprecipitation of p35 with Cdk5 before and after a light pulse.

The online version of this article includes the following figure supplement(s) for figure 2:

**Figure supplement 1.** Light-induced phosphorylation of CaMKII and control immunoprecipitation.

The CREB/CRE transcriptional pathway has been shown to be activated by calcium/calmodulin-dependent kinase II (CaMKII) and mitogen-activated protein kinase (MAPK) (*Sheng et al., 1991*; *Xing et al., 1996*; *Impey et al., 1998*). Pharmacological inhibition of CaMKII but not of MAPK affected light-induced phase delays in hamsters (*Yokota et al., 2001*). Therefore, we tested whether phosphorylated CaMKII (p-CaMKII) is affected by the knock-down of *Cdk5* in the SCN of mice. We observed that p-CaMKII presence (alpha isoform) in the cytoplasm of SCN cells increased after light at ZT14 compared to no light in control animals (*Figure 3b*, left panels). In sh*Cdk5* SCN, however, p-CaMKII was already present before the light pulse in significantly higher levels than controls (*Figure 3b*, control *Figure 3—figure supplement 1d*). This result indicates that *Cdk5* is gating the phosphorylation of CaMKII alpha.

CaMKII has been shown to shuttle Ca$^{2+}$/calmodulin (Ca$^{2+}$/CAM) to the nucleus to trigger CREB phosphorylation and gene expression (*Ma et al., 2014*). Therefore, we investigated whether CAM localization was influenced by a light pulse and whether *Cdk5* plays a role in this process. We observed that in control animals, CAM was distributed evenly in the cytoplasm of cells in SCN tissue before a light pulse. However, after the light pulse, it was localized around the nuclei (*Figure 3c*). Interestingly, in the SCN of sh*Cdk5* animals, CAM was already localized around the nuclei before the light administration and remained there after the light pulse, suggesting that *Cdk5* is gating CAM localization in the cell.

Once delivered to the nucleus, Ca$^{2+}$/CAM triggers a highly cooperative activation of the nuclear CaMK cascade, including CaMKIV, to rapidly phosphorylate CREB for the transcription of target genes (*Ma et al., 2014*; *Matthews et al., 1994*). Therefore, we tested whether a light pulse affected the phosphorylation of CaMKIV and whether this was influenced by *Cdk5*. In control animals, we detected p-CaMKIV to be strongly present in the SCN after, but not before, a light pulse (*Figure 3d*, control *Figure 3—figure supplement 1e*). In sh*Cdk5* SCN, p-CaMKIV was always detectable, independent of the light pulse (*Figure 3d*). This indicated that Cdk5 was gating phosphorylation of CaMKIV.

Calcium entry is regulated by channels, such as T-type VGCC, which are involved in phase delays (*Kim et al., 2005*). Previous reports show that Cdk5 directly or indirectly can phosphorylate Cav3.1 in vitro (*Calderón-Rivera et al., 2015*). Thus, we looked at the influence of light and *Cdk5* on the T-type channel Cav3.1 using immunohistochemical staining. We observed that the level of Cav3.1 protein was significantly increased on the surface of SCN cells after the light pulse (*Figure 3e*, blue bars). This suggests that light inhibits internalization and degradation of this channel. Interestingly, in the *Cdk5*-depleted SCN cells, Cav3.1 staining was already high on the cell surface before the light signal (*Figure 3e*, red bars). We observed no difference in the Cav3.1 signal between SCN samples obtained from sh*Cdk5* mice before and after the light pulse (*Figure 3e*, red bars), suggesting that *Cdk5* may be directly or indirectly involved in the regulation of Cav3.1 localization. This is consistent with previously described effects of Cdk5 on the cellular localization of other receptors such as the D2 and TRPV1 receptors (*Liu et al., 2019*; *Jeong et al., 2013*).

## Cdk5 modulates neuronal activity in response to light at ZT14

Neuronal activity in response to light at ZT14 requires calcium influx. At night, neuronal cell membranes are hyperpolarized, creating a Ca$^{2+}$ gradient. A light stimulus at night promotes membrane depolarization and VGCC activation, which evokes a Ca$^{2+}$ influx into SCN neurons, ultimately changing the phase of the circadian clock (*Colwell, 2001*; *Irwin and Allen, 2007*). Our results shown in *Figure 3* indicate that Cdk5 regulates the gating between light and the CaMKII pathway, which relies on Ca$^{2+}$ availability. Thus, we tested whether Cdk5 regulated the light-mediated Ca$^{2+}$ influx into SCN neurons.

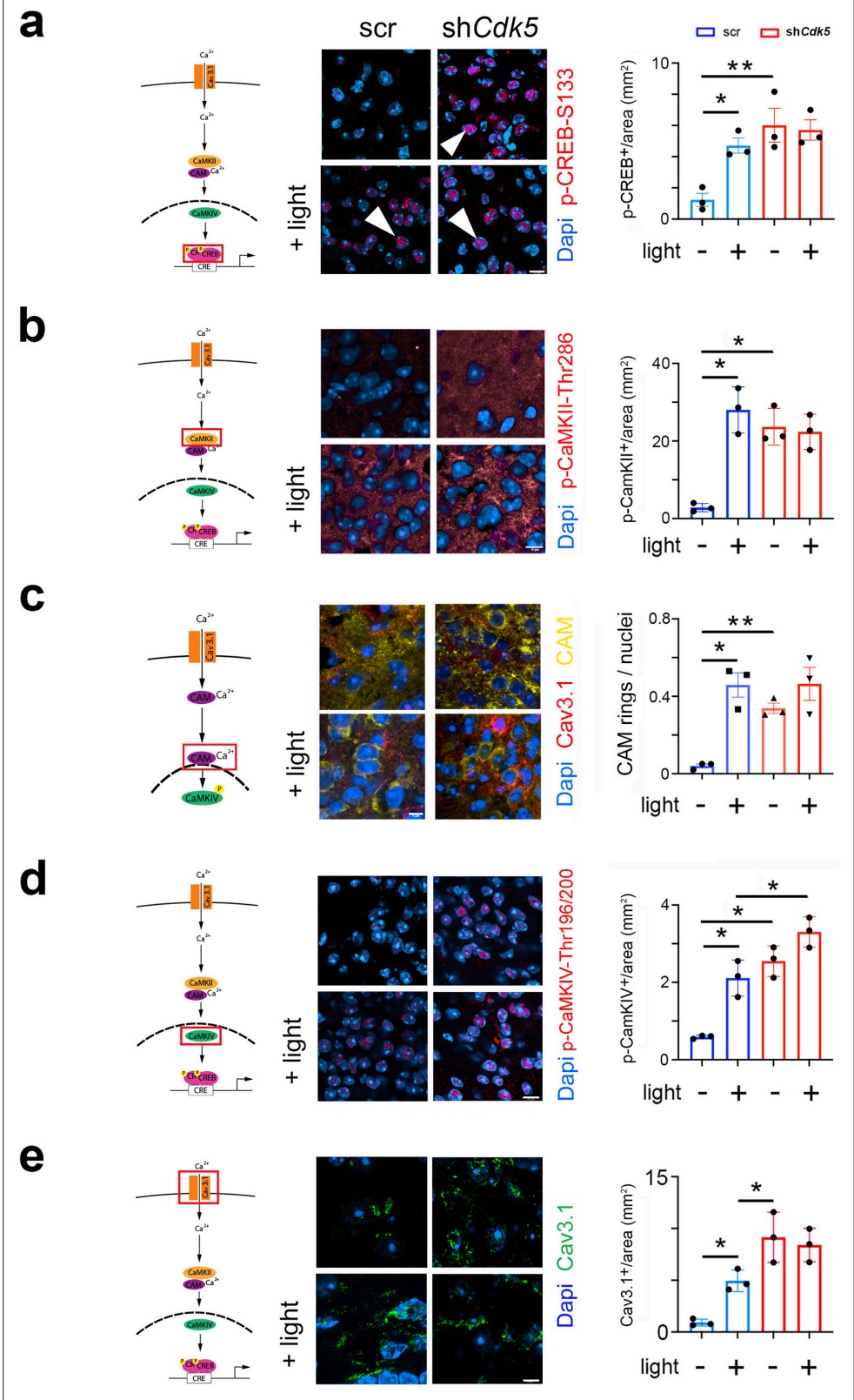

**Figure 3.** Cdk5 impacts on the cAMP response element-binding protein (CREB) signaling pathway via calcium/calmodulin-dependent kinases (CaMK). The cartoons on the left of each figure depict the CaMK pathway with the red rectangle indicating the visualization of a particular component. (**a**) Immunohistochemistry on the suprachiasmatic nucleus (SCN) of control (scr) and sh*Cdk5* mice using an antibody recognizing phospho-serine 133

*Figure 3 continued on next page*

*Figure 3 continued*

of CREB (p-CREB-S133) before and after a light pulse at zeitgeber time (ZT) 14. The red color shows p-CREB-S133 and the blue color represents DAPI-stained nuclei of SCN cells. Scale bar: 8 μm. The right panel shows the quantification of the p-CREB-S133 signal. Values are the mean ± SEM. Unpaired t-test with Welch's correction, n=3, *p<0.05. (**b**) Immunohistochemistry on the SCN of control (scr) and sh*Cdk5* mice using an antibody recognizing phosphorylated Thr286 Cam kinase II (p-CaMKII) before and after a light pulse at ZT14. The red color shows p-CaMKII and the blue color represents DAPI-stained nuclei of SCN cells. Scale bar: 8 μm. The right panel shows the quantification of the p-CaMKII signal. Values are the mean ± SEM. Unpaired t-test with Welch's correction, n=3, *p<0.05. (**c**) Translocation of calmodulin (CAM) in response to a light pulse at ZT14 in SCN neurons of control (scr) and *Cdk5* knock-down (sh*Cdk5*) animals. CAM (yellow) accumulates around the nuclei in scr controls. In sh*Cdk5* SCN neurons, this accumulation around the nuclei is already seen before the light pulse which is clearly different from scr controls. Scale bar: 7 μm The right panel shows the quantification of CAM rings. Values are the mean ± SEM. Unpaired t-test with Welch's correction, n=3, *p<0.05, **p<0.01. (**d**) Immunohistochemistry on the SCN of control (scr) and sh*Cdk5* mice using an antibody recognizing phosphorylated Thr196-200 Cam kinase IV (p-CaMKIV) before and after a light pulse at ZT14. The red color shows p-CaMKIV and the blue color represents DAPI-stained nuclei of SCN cells. Scale bar: 5 μm. The right panel shows the quantification of the p-CaMKIV signal. Values are the mean ± SEM. Unpaired t-test with Welch's correction, n=3, *p<0.05. (**e**) Immunohistochemistry on the SCN of control (scr) and sh*Cdk5* mice using an antibody recognizing the calcium channel Cav3.1 before and after a light pulse at ZT14. The green color shows Cav3.1 and the blue color represents DAPI-stained nuclei of SCN cells. Scale bar: 5 μm. The right panel shows the relative Cav3.1 signal. Values are the mean ± SEM. Unpaired t-test with Welch's correction, n=3, *p<0.05.

The online version of this article includes the following figure supplement(s) for figure 3:

**Figure supplement 1.** Lower magnification images of suprachiasmatic nucleus (SCN) sections and quantification of total cAMP response element-binding protein (CREB), CaMKII, and CaMKIV.

To this end, we employed in vivo calcium imaging to assess changes in calcium levels in the SCN in freely moving mice after 15 min of a light pulse given at ZT14. First, we injected an AAV expressing the sh*Cdk5* sequence into the SCN to silence *Cdk5*. This AAV co-expresses the calcium indicator GCaMP7 under the neuron-specific synapsin 1 promoter. As a control, we injected an AAV carrying a non-specific shRNA (scrambled sequence) instead of sh*Cdk5* (see Methods). Consistent with our previous results, the construct expressing sh*Cdk5* in the SCN produced a shortened free-running period in mice (*Figure 4—figure supplement 1a–c*). Animals injected with AAV were implanted with a chronic optical fiber placed above the SCN to allow for longitudinal imaging of GCaMP7 signals using fiber photometry. After habituation, $\Delta F/F_0$ (or the ratio of change in GCaMP7 fluorescence to the baseline fluorescence, see Methods) was recorded before and after light pulse delivery at ZT14 in both groups of mice (*Figure 4a*).

We observed an increase in calcium activity in control mice (scramble) during the second half of the 15 min of the light pulse at ZT14, which was also sustained for over 15 min after the light pulse (*Figure 4b*, black trace). In contrast, the $\Delta F/F_0$ in sh*Cdk5* mice during the second half and after the 15 min light pulse was significantly lower compared to the control animals (*Figure 4b*, red trace). This calcium activity was significantly decreased in sh*Cdk5* mice during the last 5 min of the light pulse as compared to the baseline levels (see Methods; *Figure 4c*; *Figure 4—figure supplement 1d*). Finally, mice were sacrificed, and the GFP signal was assessed by immunostaining to verify virus expression in the SCN (*Figure 4d*). The outlined circle in red indicates where the fibers were located. Taken together, these results indicate that Cdk5 modulates $Ca^{2+}$-mediated neuronal signaling.

## Cdk5 regulates the DARPP32-PKA axis

The cAMP-activated PKA signaling pathway, which leads to phosphorylation of CREB, plays a pivotal role in regulating phase delays in photic resetting (*Gillette and Mitchell, 2002*; *Sterniczuk et al., 2014*). Since the PKA signaling pathway can be induced in vivo (*Ginty et al., 1993*; *Tischkau et al., 2000*) and in vitro (*Yagita and Okamura, 2000*), we investigated whether Cdk5 could play a role in PKA-mediated CREB phosphorylation. To this end, we employed Förster resonance energy transfer (FRET), a widely used method to investigate molecular interactions between proteins such as CREB and CBP in living cells (*Brenna et al., 2021*; *Friedrich et al., 2010*).

We transfected control (wt) and *Cdk5* knock-out (*Cdk5* ko) NIH 3T3 cell lines (*Brenna et al., 2019*) with ICAP (an indicator of CREB activation due to phosphorylation) and stimulated the cells with

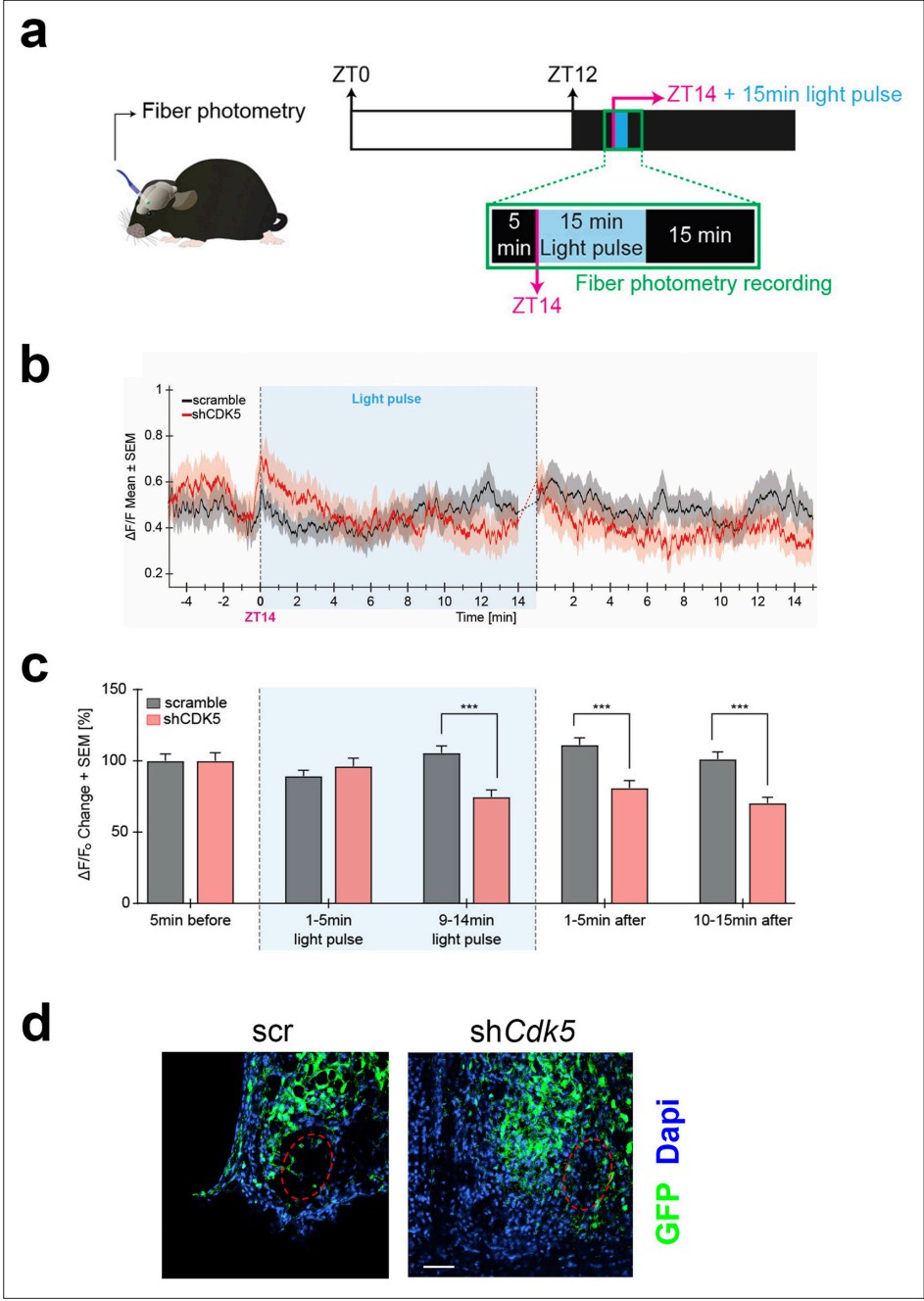

**Figure 4.** Neuronal activity in response to light at zeitgeber time (ZT) 14 is modulated by CDK5. (**a**) Illustration of the chronic optic fiber implantation in the suprachiasmatic nucleus (SCN) for fiber photometry recording in freely moving mice (left). The animals were previously infected either with AAV9-hSyn1-chl[1x(shNS)]-jGCaMP7b-WPRE-SV40p(A)(scr) or AAV9-hSyn1-chl[mouse(sh*Cdk5*)]-jGCaMP7b-WPRE-SV40p(A)(sh*Cdk5*). The experimental timeline of one trial is shown on the right. White and dark boxes represent the light and dark phase, respectively. Fiber photometry recordings were done in the 5 min before the light pulse, during the light pulse, and 15 min after. The light pulse was 15 min long and was delivered at ZT14 (blue box; dashed lines between min 14–15 are not included in the analysis). (**b**) Mean traces ± SEM of cell activity (normalized $\Delta F/F_0$) of GCaMP7b-expressing SCN neurons (black: scr, red: sh*Cdk5*) 5 min before, 15 min during, and 15 min after the light pulse (±20 s) light pulse delivered at ZT14. N=15 trials, n=5 mice/red = sh*Cdk5*, N=12 trials, n=4 mice. (**c**) Bar plot showing the percentage of $\Delta F/F_0$ changes ± SEM 5 min before the light pulse, in the first and last 5 min during the light pulse and the first and last 5 min after the light pulse. *9–14 min of light pulse*: scramble (black bar) 105.5±19.3 $\Delta F/F_0$ vs. sh*Cdk5* (red bar) 74.7±17.2 $\Delta F/F_0$. *1–5 min after light pulse*: scramble (black bar) 111.2±19.3 $\Delta F/F_0$ vs. sh*Cdk5* (red bar) 81.0±17.8 $\Delta F/$

*Figure 4 continued on next page*

*Figure 4 continued*

$F_0$. Black = scr, N=15 trials, n=5 mice/red = sh*Cdk5*, N=12 trials, n=4 mice. Bar values represent the mean ± SEM. ***p<0.001; two-way ANOVA corrected with Bonferroni post hoc test. (**d**) Photomicrograph of the expression of GCaMP7b (green) in the SCN in both control (scr, left) and experimental (sh*Cdk5*, right) animals. The red hatched oval indicates the placement of the optic fiber. Blue: DAPI, green: GFP (produced by jGCaMPP7). Scale bar: 50 µm.

The online version of this article includes the following figure supplement(s) for figure 4:

**Figure supplement 1.** GCaMP7 reporter does not affect knock-down of Cdk5 in the suprachiasmatic nucleus (SCN).

forskolin in the presence of $Ca^{2+}$. The difference in phosphorylation before and after forskolin treatment of the CREB domain in the reporter decreases the FRET signal normally between 10 and 30 min, while no difference in phosphorylation brings the FRET signal back toward baseline. We observed that the FRET signal in control cells strongly decreased between 10 and 30 min after the stimulus compared to baseline (*Figure 5a*, blue trace, the first 5 min are ignored, because they represent the diffraction of the solvent DMSO). In contrast, the FRET signal in *Cdk5* ko cells rose toward baseline after an initial decline in response to forskolin (*Figure 5a*, red trace). This indicated that Cdk5 is involved in the phosphorylation of CREB. Notably, the forskolin solvent DMSO can't stimulate CREB phosphorylation on its own (*Figure 5—figure supplement 1a*).

Previous studies have described that $Ca^{2+}$-mediated CREB transcription of target genes requires PKA activity (*Impey et al., 1998*). However, it is not clear whether there is a parallel (synergistic) relationship between PKA and $Ca^{2+}$ signaling pathways or whether they are sequentially dependent on each other (*Figure 5b*, cartoon model). To address this question, we performed the following FRET experiment. NIH 3T3 cells were stimulated with forskolin in the presence of $Ca^{2+}$ with EGTA ($Ca^{2+}$ chelator) (*Figure 5b*, red line), without EGTA (*Figure 5b*, blue line) or completely depleted of $Ca^{2+}$ (*Figure 5b*, orange line). We observed that under normal conditions the FRET signal decreased, comparable to the signal seen in *Figure 5a*, indicating higher Ser-133 KID phosphorylation compared to the baseline (*Figure 5b*, blue line). When we added EGTA (removing $Ca^{2+}$), the FRET signal increased to the baseline level after forskolin treatment (*Figure 5b*, red line). The cells depleted of $Ca^{2+}$ were also not responsive to the forskolin stimulus, as the FRET signal moved toward the baseline level within 30 min (*Figure 5b*, orange line). Together, our results indicate that CREB phosphorylation is modulated by Cdk5 via $Ca^{2+}$ signaling, as suggested in *Figure 3*. Interestingly, PKA did not appear to directly phosphorylate CREB, as CREB did not pull down p-PKA in an immunoprecipitation experiment. In contrast, p-CaMKIV did interact with CREB (*Figure 5—figure supplement 1b and c*), suggesting that CREB is most likely phosphorylated by CaMKIV, which is probably indirectly regulated by PKA activity.

Next, we aimed to investigate what the possible pathway could be through which PKA regulates CaMKIV. Previous studies have shown that Cdk5 regulates PKA activity via DARPP32 (*Svenningsson et al., 2004*; *Figure 5c*). Therefore, we asked whether DARPP32 phosphorylation was light-dependent and whether Cdk5 would modulate this process. We sacrificed mice either receiving a light pulse at ZT14 or no light. Cryosections containing the SCN were stained with an antibody recognizing phosphorylated Thr-75 (pThr-75) of DARPP32. We observed that DARPP32 is highly phosphorylated at ZT14, with the light signal significantly reducing the phosphorylation levels in the cytoplasm and nuclei (*Figure 5d*, blue bars; *Figure 5—figure supplement 1d and e*). In contrast, silencing of *Cdk5* led to a dramatic decrease in the pThr-75 signal in the cytoplasm and nuclei of SCN cells at ZT14, and light did not have an effect (*Figure 5d*, red bars, *Figure 5—figure supplement 1e*). These observations are consistent with the view that Cdk5 phosphorylates DARPP32 and that light inhibits this process.

Non-phosphorylated DARPP32 promotes PKA activity, characterized by phosphorylation at Thr-197 in the catalytic site of PKA (*Yonemoto et al., 1993*; *Cauthron et al., 1998*). Therefore, we asked whether decreased levels of p-DARPP32 after the light stimulus at ZT14 could inversely correlate with the phosphorylation state of PKA. We performed immunostaining on coronal brain sections containing the SCN using an antibody recognizing the phosphorylated Thr-197 of PKA. We observed that PKA phosphorylation significantly increased after the light pulse in the SCN tissue obtained from control (scr) mice (*Figure 5e*, right panel, blue bars). However, in SCN from sh*Cdk5* mice, the phosphorylation level was already elevated before the light pulse compared to scr control (*Figure 5e*, left panels, top micrographs), and it was also sustained after the light pulse (*Figure 5e*, left panels, bottom

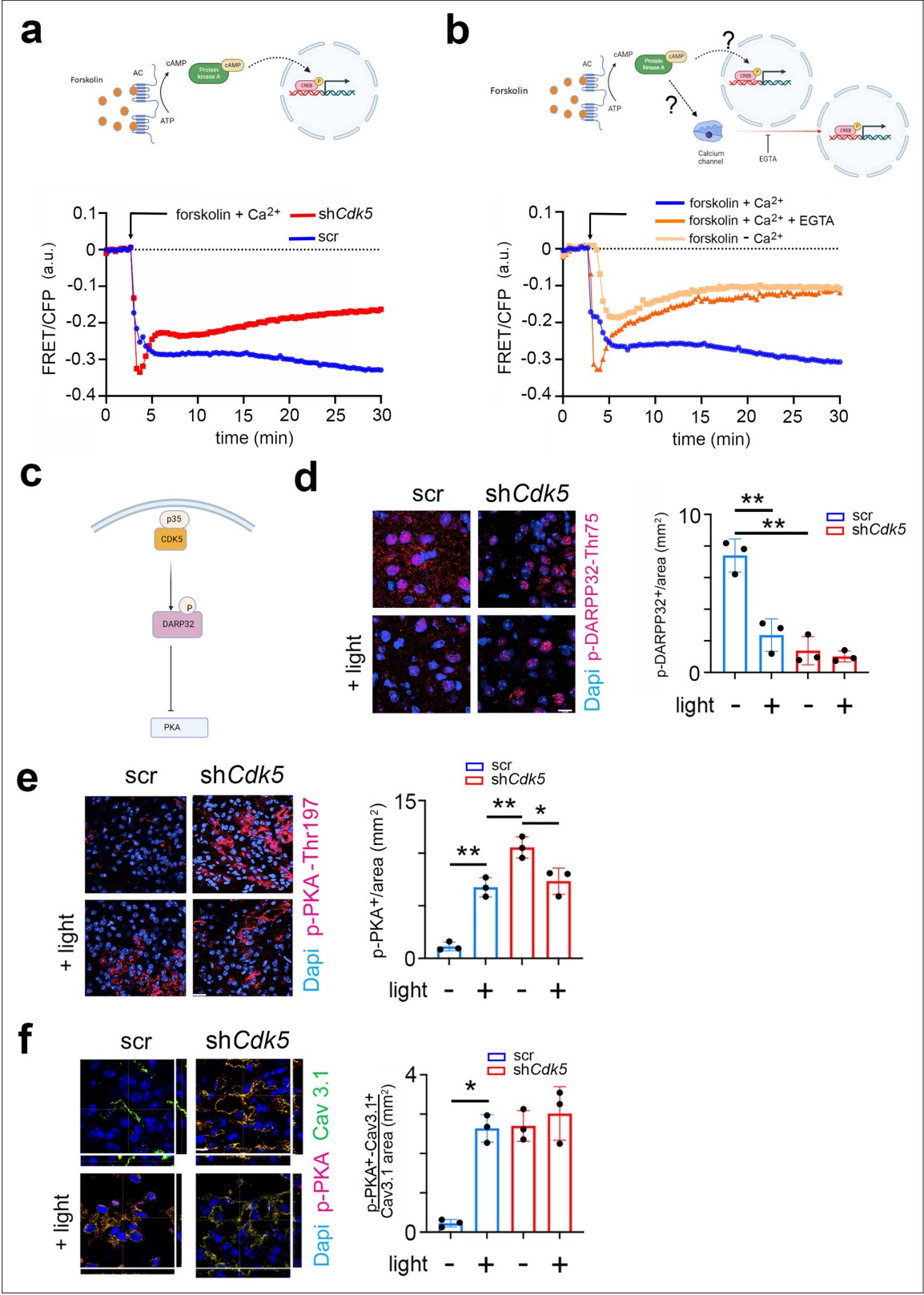

**Figure 5.** CDK5 regulates PKA phosphorylation via DARPP32 phosphorylation. (**a**) Top: Scheme of the forskolin-PKA-CREB signaling pathway. Bottom: Förster resonance energy transfer (FRET)/CFP signal ratio changes in response to forskolin treatment in NIH 3T3 cells transfected with either a scr control (blue) or sh*Cdk5* (red) expression construct. Values are the mean ± SD. Two-way ANOVA revealed a significant difference between the curves, n=3, ****p<0.0001. (**b**) Top: Scheme of the forskolin-PKA-CREB signaling pathway and calcium signaling. Bottom: FRET/CFP signal ratio changes in

*Figure 5 continued on next page*

*Figure 5 continued*

response to forskolin treatment in NIH 3T3 cells with the addition of $Ca^{2+}$ (blue), without the addition of $Ca^{2+}$ (salmon colored), and with the addition of $Ca^{2+}$ and EGTA (orange). Values are the mean ± SD. Two-way ANOVA revealed a significant difference between the gray and blue/orange curves, n=3, ****p<0.0001. (**c**) Scheme of CDK5-DARPP32-PKA pathway. (**d**) Immunohistochemistry on the suprachiasmatic nucleus (SCN) of control (scr) and sh*Cdk5* mice using an antibody recognizing phosphorylated Thr-75 of DARPP32 (p-DARPP32) before and after a light pulse at zeitgeber time (ZT) 14. The red color shows p-DARPP32 and the blue color represents DAPI-stained nuclei of SCN cells. Scale bar: 10 μm. The right panel shows the quantification of the p-DARPP32 signal. Values are the mean ± SEM. Unpaired t-test with Welch's correction, n=3, **p<0.01. (**e**) Immunohistochemistry on the SCN of control (scr) and sh*Cdk5* mice using an antibody recognizing phosphorylated Thr-197 of PKA (p-PKA) before and after a light pulse at ZT14. The red color shows p-PKA and the blue color represents DAPI-stained nuclei of SCN cells. Scale bar: 20 μm. The right panel shows the quantification of the p-PKA signal. Values are the mean ± SEM. Unpaired t-test with Welch's correction, n=3, *p<0.05, **p<0.01. (**f**) Immunohistochemistry on the SCN of control (scr) and sh*Cdk5* mice using an antibody recognizing phosphorylated Thr-197 of PKA (p-PKA) and Cav3.1 before and after a light pulse at ZT14. The red color shows p-PKA, the green color Cav3.1, and the blue color represents DAPI-stained nuclei of SCN cells. The yellow color signifies the co-localization of PKA and Cav3.1. The stripes on the left and bottom of each micrograph show the z-stacks to confirm co-localization. Scale bar: 10 μm. The right panel shows the quantification of relative p-PKA/Cav3.1. Values are the mean ± SEM. Unpaired t-test with Welch's correction, n=3, *p<0.05.

The online version of this article includes the following figure supplement(s) for figure 5:

**Figure supplement 1.** Solvent control for Förster resonance energy transfer (FRET) experiment.

micrographs, right panel, red bars). Our results indicate that Cdk5 gates PKA phosphorylation induced by the light pulse at ZT14. Many observations indicate that active PKA can stimulate the $Ca^{2+}$ influx through Cav3 T-type voltage-gated channels, including Cav3.1 (*Chemin et al., 2007*; *Harraz and Welsh, 2013*). The molecular mechanism normally requires physical interaction between the channel and PKA, followed by phosphorylation, which influences the gating properties (*Lory et al., 2020*). Therefore, we performed a co-immunostaining in the same SCN sections collected before (*Figure 3*) to detect both Cav3.1 and phospho-PKA (the active form). We observed that the co-localization between Cav3.1 and phospho-PKA dramatically increased after the light pulse in the SCN tissue of control (scr) mice (*Figure 5f*, scr left panels yellow color, and blue bars in the right panel). Interestingly, the co-localization level of the two proteins was already high in the sh*Cdk5* SCN tissue before the light pulse, compared to controls (*Figure 5f* scramble vs. sh*Cdk5*, left panel, top micrographs). The co-localization level between Cav3.1 and phospho-PKA in the sh*Cdk5* tissues was not influenced by the light pulse (*Figure 5f*, right panel, red bars). Altogether our results suggest that Cdk5 gates the PKA-Cav3.1 interaction in response to the light signal at ZT14 in an indirect way via DARPP32.

## Cdk5 affects light-induced gene expression

Light perceived in the dark period leads not only to phase shifts but also induces immediate early genes and certain clock genes in the SCN (*Rusak et al., 1990*; *Albrecht et al., 1997*; *Shigeyoshi et al., 1997*; *Kornhauser et al., 1990*; *Honma et al., 2002*). This process involves the PKA-CaMK-CREB signaling pathway (reviewed in *Golombek and Rosenstein, 2010*). Therefore, we investigated whether Cdk5 is involved in the signal transduction process to induce immediate early genes and clock genes in the SCN in response to light. To this end, we performed a time-course profile of light-induced genes and immediate early genes. We collected SCN from mice that received a nocturnal light pulse at ZT14 at different time points over 2 hr (*Figure 6*).

In agreement with previous studies, *Per1* and *Dec1* mRNA expression was induced by light, peaking at 1 hr after the stimulus. Conversely, *Per2* and *Dec2* mRNA expression was not affected by the light pulse at ZT14 (*Figure 6a–d*, blue bars) (*Shigeyoshi et al., 1997*; *Honma et al., 2002*; *Olejniczak et al., 2021*). Knock-down of *Cdk5* abolished this light-driven *Per1* and *Dec1* gene induction (*Figure 6a and c*, red bars), indicating the involvement of *Cdk5* in the light-driven activation process of these clock genes. As previously reported, expression of the clock gene *Bmal1* was not light-inducible (*Brenna et al., 2019*; *von Gall et al., 2003*) and was not affected by sh*Cdk5* (*Figure 6e*). The injection of the control scr and sh*Cdk5* constructs was successful, as demonstrated by the expression of *eGFP* mRNA in the analyzed SCN (*Figure 6f*).

Interestingly, the knock-down of *Cdk5* did not affect light-mediated induction of *Fos* expression, which peaked at 0.5 hr after the light pulse (*Figure 6g*). In contrast, *Egr1*, another immediate early gene involved in synaptic plasticity, learning, and memory (*Li et al., 2005*), was light-inducible in control but not in sh*Cdk5* animals (*Figure 6h*). This suggests that the immediate early gene *Fos* is

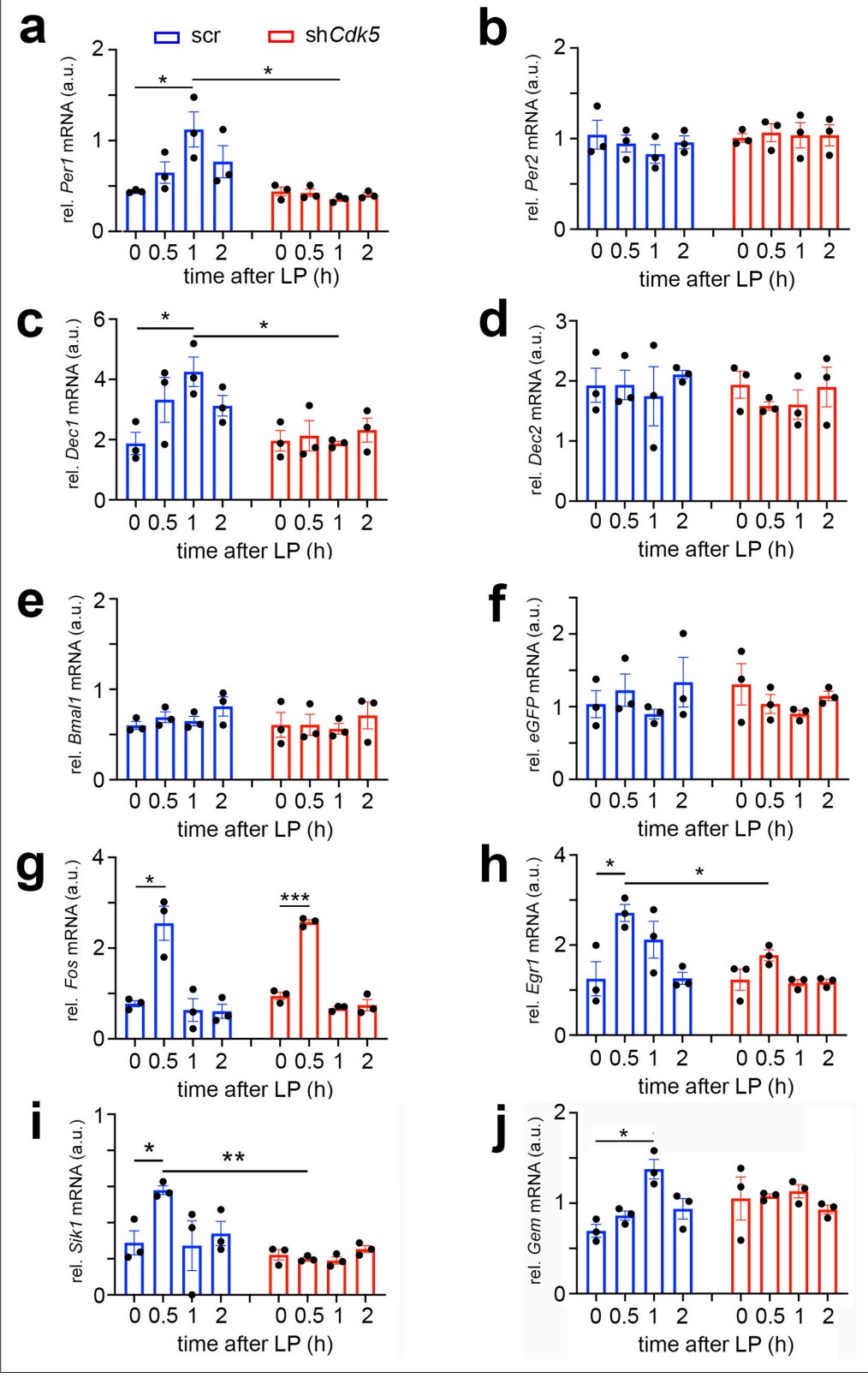

**Figure 6.** Cdk5 regulates light-induced gene expression in the suprachiasmatic nucleus (SCN) of some clock genes. Relative mRNA values are represented as blue bars for scr control animals and as red bars for sh*Cdk5* mice. The values were determined 0, 0.5, 1, and 2 hr after a light pulse (LP) given at zeitgeber time (ZT) 14. (**a**) Induction of *Per1* mRNA expression by light with a maximum of 1 hr after light in scr control animals. In contrast, *Per1*

*Figure 6 continued on next page*

*Figure 6 continued*

is not induced in sh*Cdk5* SCN. Scr: 0 hr: 0.44±0.01, 0.5 hr: 0.65±0.12, 1 hr: 1.12±0.19, 2 hr: 0.44±0.01; sh*Cdk5*: 0 hr: 0.44±0.01, 0.5 hr: 0.44±0.01, 1 hr: 0.44±0.01, 2 hr: 0.77±0.18. Values are the mean ± SEM. Unpaired t-test, n=3, *p<0.05. (**b**) *Per2* mRNA expression is not induced by light neither in scr controls nor in sh*Cdk5* animals. Scr: 0 hr: 1.04±0.16, 0.5 hr: 0.95±0.10, 1 hr: 0.83±0.10, 2 hr: 0.96±0.07; sh*Cdk5*: 0 hr: 1.01±0.05, 0.5 hr: 1.07±0.10, 1 hr: 1.04±0.14, 2 hr: 1.04±0.12. Values are the mean ± SEM. Unpaired t-test, n=3. (**c**) Induction of *Dec1* mRNA expression by light with a maximum at 1 hr after light in scr control animals. In contrast, *Dec1* is not induced in sh*Cdk5* SCN. Scr: 0 hr: 1.87±0.37, 0.5 hr: 3.32±0.75, 1 hr: 4.25±0.49, 2 hr: 3.13±0.34; sh*Cdk5*: 0 hr: 1.96±0.34, 0.5 hr: 2.13±0.51, 1 hr: 1.87±0.07, 2 hr: 2.32±0.40. Values are the mean ± SEM. Unpaired t-test, n=3, *p<0.05. (**d**) *Dec2* mRNA expression is not induced by light neither in scr controls nor in sh*Cdk5* animals. Scr: 0 hr: 1.93±0.29, 0.5 hr: 1.94±0.25, 1 hr: 1.75±0.49, 2 hr: 2.11±0.07; sh*Cdk5*: 0 hr: 1.94±0.23, 0.5 hr: 1.58±0.07, 1 hr: 1.61±0.25, 2 hr: 1.90±0.33. Values are the mean ± SEM. Unpaired t-test, n=3. (**e**) *Bmal1* mRNA expression is not induced by light in the SCN of scr control and sh*Cdk5* animals. Scr: 0 hr: 0.60±0.04, 0.5 hr: 0.69±0.06, 1 hr: 0.65±0.05, 2 hr: 0.81±0.12; sh*Cdk5*: 0 hr: 0.61±0.14, 0.5 hr: 0.61±0.12, 1 hr: 0.56±0.06, 2 hr: 0.71±0.15. Values are the mean ± SEM. Unpaired t-test, n=3. (**f**) *eGFP* mRNA expression is detected in the SCN of scr control and sh*Cdk5* animals demonstrating proper injection of expression constructs (scr: ssAAV-9/2-hSyn1-chl[1x(shNS)]-EGFP-WPRE-SV40p(A), sh*Cdk5*: ssAAV-9/2-hSyn1-chl[mouse(sh*Cdk5*)]-EGFP-WPRE-SV40p(A)). Scr: 0 hr: 1.04±0.19, 0.5 hr: 1.23±0.22, 1 hr: 0.90±0.07, 2 hr: 1.34±0.34; sh*Cdk5*: 0 hr: 1.31±0.28, 0.5 hr: 1.04±0.13, 1 hr: 0.90±0.05, 2 hr: 1.15±0.06. Values are the mean ± SEM. Unpaired t-test, n=3. (**g**) Induction of *Fos* mRNA 0.5 hr after the light pulse in both scr controls and sh*Cdk5* SCN. Scr: 0 hr: 0.77±0.07, 0.5 hr: 2.55±0.38, 1 hr: 0.64±0.25, 2 hr: 0.61±0.15; sh*Cdk5*: 0 hr: 0.95±0.08, 0.5 hr: 2.57±0.05, 1 hr: 0.68±0.04, 2 hr: 0.74±0.12. Values are the mean ± SEM. Unpaired t-test, n=3, *p<0.05, ***p<0.001. (**h**) Induction of *Egr1* mRNA 0.5 hr after the light pulse in scr control but not sh*Cdk5* SCN. Scr: 0 hr: 1.25±0.38, 0.5 hr: 2.71±0.19, 1 hr: 2.12±0.41, 2 hr: 1.26±0.13; sh*Cdk5*: 0 hr: 1.23±0.24, 0.5 hr: 1.77±0.13, 1 hr: 1.16±0.08, 2 hr: 1.18±0.06. Values are the mean ± SEM. Unpaired t-test, n=3, *p<0.05. (**i**) *Sik1* mRNA expression is induced by light in the SCN of scr control but not sh*Cdk5* animals. Scr: 0 hr: 0.29±0.07, 0.5 hr: 0.58±0.02, 1 hr: 0.27±0.14, 2 hr: 0.34±0.07; sh*Cdk5*: 0 hr: 0.22±0.03, 0.5 hr: 0.20±0.01, 1 hr: 0.19±0.02, 2 hr: 0.25±0.02. Values are the mean ± SEM. Unpaired t-test, n=3, *p<0.05, **p<0.01. (**j**) *Gem* mRNA expression is induced by light in the SCN of scr control but not sh*Cdk5* animals. Scr: 0 hr: 0.70±0.07, 0.5 hr: 0.86±0.05, 1 hr: 1.38±0.11, 2 hr: 0.94±0.11; sh*Cdk5*: 0 hr: 1.05±0.24, 0.5 hr: 1.08±0.03, 1 hr: 1.13±0.07, 2 hr: 0.93±0.05. Values are the mean ± SEM. Unpaired t-test, n=3, *p<0.05.

The online version of this article includes the following figure supplement(s) for figure 6:

**Figure supplement 1.** *Vip*, *Per1*, *Dec1* levels in the suprachiasmatic nucleus (SCN) of sh*Cdk5* and *Per2*[Brdm1] mice and *Per1, 2* and *Fos* gene expression in NIH 3T3 cells.

---

regulated by a different mechanism compared to *Egr1* and the clock genes *Per1* and *Dec1* in response to a light stimulus at ZT14.

Vasoactive intestinal polypeptide (VIP) has been described to play a role in phase-shifting the SCN clock (**Reed et al., 2001**). Furthermore, the light-induced expression of clock genes is localized in VIP-positive cells in the SCN, which are essential for clock resetting (**Jones et al., 2018**). Therefore, we tested whether *Vip* gene expression is affected by sh*Cdk5*. We observed that a light pulse did not significantly induce *Vip* expression in the SCN, nor did sh*Cdk5* affect its general expression (**Figure 6—figure supplement 1a**). This suggests that *Cdk5* does not regulate *Vip* expression and modulate phase shifts via VIP.

Salt-inducible kinase 1 (*Sik1*) is involved in the regulation of the magnitude and duration of phase shifts by acting as a suppressor of the effects of light on the clock (**Jagannath et al., 2013**). Therefore, we tested how a light pulse affected *Sik1* expression in the SCN and whether *Cdk5* might play a role in its regulation. We observed that *Sik1* was significantly induced by a light pulse in the SCN of control mice after 0.5 hr. However, the knock-down of *Cdk5* abolished this induction (**Figure 6i**). This suggests that *Cdk5* modulates *Sik1* expression to regulate the magnitude of the behavioral response to light.

The light-inducible small G-protein Gem limits the circadian clock phase-shift magnitude by inhibiting voltage-dependent calcium channels (**Matsuo et al., 2022**). We tested whether a light pulse affected *Gem* expression in the SCN and whether this involved Cdk5. We observed that *Gem* was significantly induced by light 1 hr after light administration (**Figure 6j**, blue bars). Interestingly, knockdown of *Cdk5* abolished this induction (**Figure 6i**, red bars), but *Gem* levels seemed to be slightly elevated already before light administration (**Figure 6i**, time point 0). This indicates that Cdk5 influences light-induced *Gem* expression and may also affect basal *Gem* expression before the light pulse. Similar results for light-induced gene expression in sh*Cdk5* SCN were observed in SCN of *Per2*[Brdm1] mutant mice (**Figure 6—figure supplement 1b and c**).

Phase shifts of the circadian clock can also be studied in cell cultures using forskolin instead of light as a stimulus (*Yagita and Okamura, 2000*). In accordance with our in vivo experiments (*Figure 6*), expression of *Per1* but not *Per2* mRNA was induced in synchronized NIH 3T3 fibroblast cells after forskolin treatment (*Figure 6—figure supplement 1d and e*, blue bars). Comparable to the experiments in the SCN, *Per1* induction was abolished in *Cdk5* ko cells (*Figure 6—figure supplement 1d*). In contrast, *Fos* mRNA induction was not affected in *Cdk5* ko cells (*Figure 6—figure supplement 1f*), consistent with our observations in the SCN (*Figure 6g*).

Collectively, our expression data provide evidence that *Cdk5* regulates light- and forskolin-mediated expression of genes critical for the regulation of phase delays of the circadian clock. Immediate early genes, such as *Egr1*, are regulated in a similar manner, whereas others, such as *Fos*, are regulated by a different mechanism not involving *Cdk5*.

## Discussion

In this study, we investigated the role of Cdk5 in rapid phase shifts of the circadian clock. We found that Cdk5 activity is regulated by light and that Cdk5 is necessary for phase delays but not phase advances. We identified Cdk5 to play a major role in the modulation of $Ca^{2+}$ levels and gating of the PKA-CaMK-CREB signaling pathway, coordinating it with the presence of PER2 in the nucleus of SCN cells.

In a previous study, we identified the protein kinase Cdk5 to regulate the phosphorylation and nuclear localization of the clock protein PER2 (*Brenna et al., 2019*). Because PER2 and protein kinases are involved in the photic signaling mechanism of clock phase adaptation (*Brenna et al., 2021*; *Albrecht et al., 2001*; *Golombek and Rosenstein, 2010*; *Alessandro et al., 2019*; *Ashton et al., 2022*), we tested the involvement of Cdk5 in this process. The phenotype of *Cdk5* knockdown (sh*Cdk5*) in the SCN of mice resembled the phenotypes observed in *Per2* mutant (*Per2^{Brdm1}*) and neuronal *Per2* knock-out (n*Per2* ko) mice. ShCdk5, as well as *Per2^{Brdm1}* and n*Per2* ko animals, showed strongly reduced phase delays in response to a short light pulse given at ZT14 (*Figure 1a and c*; *Albrecht et al., 2001*) or CT14 (*Figure 1d and f*; *Spoelstra et al., 2004*; *Chavan et al., 2016*). These mouse lines displayed a shortened period consistent with previous observations (*Figure 1b and e*; *Brenna et al., 2019*; *Zheng et al., 1999*). Our results indicate that Cdk5 is not only involved in the regulation of the circadian clock mechanism via nuclear localization of PER2 but also plays an important role in the molecular mechanism that leads to a delay of clock phase in response to a light pulse in the early dark phase or early subjective night.

Since Cdk5 mediates the effects of light at the behavioral (*Figure 1*) level, we tested the influence of light on Cdk5 protein accumulation and kinase activity in the SCN at ZT14 (*Figure 2*). We observed no change in the protein accumulation of Cdk5. On the other hand, Cdk5 kinase activity was reduced in the SCN after a light pulse at ZT14 (*Figure 2d and e*), which was surprising in the context of increased p35 levels (*Figure 2a and c*) and augmented PKA phosphorylation (*Figure 2a and b*). However, this observation is in line with what we previously reported, where we demonstrated that Cdk5 kinase activity was low during the light phase and higher during the dark phase (*Brenna et al., 2019*). It appeared, however, that p35 was not interacting with Cdk5 after light at ZT14 (*Figure 2f*). Additional interactions of Cdk5 with unknown proteins may also be lost (*Figure 2—figure supplement 1e*). These observations suggest that Cdk5 was most likely modified in response to light leading to loss of interaction with p35 and other proteins. Ser159 of Cdk5 mediates the specificity of the Cdk5-p35 interaction (*Tarricone et al., 2001*), and therefore, phosphorylation of this site by an unknown kinase may mediate the loss of Cdk5 activity. Several additional phosphorylation sites in Cdk5 have been identified, of which phosphorylation of S47 renders Cdk5 inactive (*Roach et al., 2018*). Which one of the phosphorylation sites in Cdk5 is modulated by light and what additional interactors may be involved in this process remains to be established.

Light in the early portion of the dark phase elicits phase delays, which involve T-type calcium channels, PKA signaling, and $Ca^{2+}$ signaling, ending in the phosphorylation of CREB (reviewed in *Golombek and Rosenstein, 2010*). We observed that in sh*Cdk5* mice, CREB was already phosphorylated in the absence of light, although the total protein amount did not change (*Figure 3a*, *Figure 3—figure supplement 1a and b*). Similarly, CaMKII and CaMKIV were shown to be phosphorylated and, therefore, activated only after the light pulse in control animals (*Figure 3b and d*). Conversely, these kinases were highly phosphorylated in a light-independent manner in the SCN of sh*Cdk5* animals

(*Figure 3b and d*), indicating that Cdk5 had a suppressive function on the phosphorylation of CaMKII and CaMKIV.

A stimulus can promote calmodulin (CAM) involving CaMKII gamma to translocate from calcium channels to the nucleus to promote CaMKIV phosphorylation and activation (*Ma et al., 2014*). Unexpectedly, we observed that a light stimulus can have a similar but distinct effect on CAM in SCN cells (*Figure 3c*). CaMKII alpha was phosphorylated after a light pulse at ZT14, which led to perinuclear localization of CAM in control mice, while this localization pattern was already observed in sh*Cdk5* animals independently of the light stimulus (*Figure 3c*). In control mice that received no light, CAM showed a diffuse expression pattern similar to the T-type calcium channel Cav3.1 at ZT14.

Interestingly, this light-driven localization pattern was echoed by the change in cellular distribution of the T-type calcium channel Cav3.1 known as internalization/externalization. Again, the presence of Cdk5 suppressed the localization of this channel to the cell membrane in the absence of light, with light allowing localization to the cell membrane (*Figure 3e*). This observation is reminiscent of investigations described previously in which Cdk5 appeared to play an important role in channel translocation (*Oberegelsbacher et al., 2011*; *Katz et al., 2017*) as well as in receptor translocation (*Liu et al., 2019*; *Jeong et al., 2013*). Thus, our findings are in accordance with the view that Cdk5 plays a crucial role in light stimulus-driven cell dynamics.

Calcium plays an important role in circadian and phase-shifting biology (*Colwell, 2001*). Circadian calcium fluxes in the cytosol of SCN neurons have been demonstrated in vitro (*Brancaccio et al., 2013*), and they change rapidly as a response to light perceived by the retina (*Irwin and Allen, 2007*). We performed in vivo live imaging to detect $Ca^{2+}$ levels in the SCN using fiber photometry with protein-based $Ca^{2+}$ indicators such as GCaMP (*Zhang et al., 2023*). With this approach, we observed that calcium fluxes in the SCN of control mice increased during and after a light pulse, but this change was significantly dampened in sh*Cdk5* animals (*Figure 4b and c*). Interestingly, although the $Ca^{2+}$ influx was generally reduced in the SCN of sh*Cdk5* mice, we observed random $Ca^{2+}$ activity, which was independent of any light stimulus. These transients were observed also at the beginning of ZT14, before the light pulse (*Figure 4b and c*). These results may indicate the presence of a calcium leak reminiscent of the already active phosphorylation cascade observed in the sh*Cdk5* SCN in the absence of light (*Figure 3*). We do not know, however, whether internal calcium stores involving ryanodine receptors (*Ding et al., 1998*) are altered by Cdk5 as well and how this would contribute to the observed phenotypes.

The PKA signaling pathway is involved in the resetting of the circadian phase (reviewed in *Gillette and Mitchell, 2002*). Interference with PKA activation in the early subjective night led to reduced phase delay responses as observed in vitro in the SCN (*Tischkau et al., 2000*). Here, we find that Cdk5 plays an inhibitory role in PKA phosphorylation and activation. The FRET approach shows that in cells the lack of Cdk5 makes cells unresponsive to forskolin (*Figure 5a*), an agent known to mitigate phase shifts in cells via PKA (*Yagita and Okamura, 2000*). Interestingly, PKA appears to influence phase shifts and CREB phosphorylation indirectly via a $Ca^{2+}$-dependent mechanism (*Figure 5b*) with phosphorylated CaMKIV being the kinase that phosphorylates CREB (*Figure 5—figure supplement 1b and c*). This observation is in contrast with previous studies that suggested a direct phosphorylation of CREB by PKA (*Ginty et al., 1993*; *Tischkau et al., 2000*). However, p-PKA is mostly located in the cytoplasm (*Figure 5e*) while p-CaMKIV is in the nuclei (*Figure 3d*). Furthermore, our experiments indicate that CREB did not interact with p-PKA but did with p-CaMKIV (*Figure 5—figure supplement 1b, c*), supporting the notion that PKA regulates CREB phosphorylation indirectly via CaMKIV in the SCN.

Because we observed that PKA was already phosphorylated in the dark when Cdk5 was silenced (*Figure 5e*), we asked how Cdk5 could negatively regulate PKA phosphorylation. A previous study described that Cdk5 can phosphorylate DARPP32 to suppress PKA activity (*Bibb et al., 1999*). Furthermore, *Darpp-32* ko mice show attenuated phase delays (*Yan et al., 2006*) resembling sh*Cdk5* mice (*Figure 1*). In accordance with these studies, we found an inverse correlation between p-DARPP32 (*Figure 5d*) and p-PKA (*Figure 5e*), implying that Cdk5 indirectly inhibits PKA activity via DARPP32. However, phosphatases such as PP2A and calcineurin, which dephosphorylate DARPP32 including the Cdk5 phosphorylation site, may be involved in this process as well (*Girault and Nairn, 2021*). Upon light treatment and increase of $Ca^{2+}$, these phosphatases would dephosphorylate DARPP32 and thereby inactivate it, leading to PKA activation. This process may occur in parallel to the Cdk5

regulation of DARPP32 contributing to a sustained activation of the light signaling pathway via PKA activation.

Our results imply that PKA action on CREB might be mediated via T-type calcium channels such as Cav3.1 (*Figure 5f*). This assumption is reasonable because PKA can phosphorylate Cav3.1 channels and increase electrical conductivity, which leads to a higher influx of $Ca^{2+}$ (*Kim et al., 2006*). To that extent, our results indicate that a higher co-localization of p-PKA with Cav3.1 is associated with an activation of the CaMK pathway and CREB phosphorylation.

Light-induced phosphorylation of CREB leads to induction of immediate early genes and clock genes (reviewed in *Golombek and Rosenstein, 2010*). Accordingly, we observed that the clock genes *Per1* and *Dec1* but not *Per2*, *Dec2,* and *Bmal1* were induced in the SCN by light at ZT14 (*Figure 6a–e*, blue bars) consistent with previous findings (*Brenna et al., 2021*; *Shigeyoshi et al., 1997*; *Honma et al., 2002*; *Olejniczak et al., 2021*). The light induction of *Per1* and *Dec1* was abolished in sh*Cdk5* animals (*Figure 6a and c*, red bars) as well as in *Per2^Brdm1^* mutant mice (*Figure 6—figure supplement 1b and c*), suggesting involvement of Cdk5 and Per2 in induction of these genes. In contrast, light induction of the immediate early gene *Fos* was neither affected in sh*Cdk5* nor *Per2^Brdm1^* SCN (*Figure 6g*, *Figure 6—figure supplement 1f*), resembling the normal *Fos* induction in *Per2* ko animals (*Brenna et al., 2021*). This indicates that the light signaling mechanism for *Fos* induction is different from the one mediating induction of *Per1* and *Dec1*. Interestingly, however, the light-inducible genes *Sik1* and *Gem*, which are involved in limiting the effects of light on the clock (*Jagannath et al., 2013*; *Matsuo et al., 2022*) were not light-inducible in sh*Cdk5* animals (*Figure 6i and j*) supporting the view that the factors that drive (*Per1*, *Dec1*) or limit (*Sik1*, *Gem*) the effects of light on the clock are regulated by the same mechanism. Interestingly, neither lack of *Per1*, *Dec1*(*Albrecht et al., 2001*; *Rossner et al., 2008*) nor *Sik1* or *Gem* (*Jagannath et al., 2013*; *Matsuo et al., 2022*) alone abolish phase delays. Of note is that lack of *Fos* or *Egr1* did not affect phase delays either (*Honrado et al., 1996*; *Riedel et al., 2018*). Furthermore, the neuropeptide VIP, which is important in circadian light responses (*Jones et al., 2018*), was not inducible by a light pulse at ZT14 in control as well as sh*Cdk5* animals (*Figure 6—figure supplement 1a*), indicating that Cdk5 acts upstream of VIP signaling. Overall, the present data suggest that Cdk5 not only regulates the light-sensitive PKA-CaMK-CREB signaling pathway but ultimately also affects gene expression. For the transcriptional activation of those genes, nuclear PER2 protein is necessary, which is regulated by Cdk5 (*Brenna et al., 2021*; *Brenna et al., 2019*). The combination of lack of induction of many genes in the Cdk5-regulated pathway is responsible for the manifestation of rapid behavioral phase delays.

Based on this and our previous studies, we propose the following molecular model for light-mediated phase delays (*Figure 7*). The model is divided into two parts. One part describes the state before the light pulse, and the second part the mechanism after the light pulse. The state before the light pulse (ZT12–14) is depicted in *Figure 7a*. As reported previously, Cdk5 is active right after dark onset (*Brenna et al., 2019*), depicted as the active Cdk5/p35 complex (blue). This has two consequences: (1) PER2 (red) is phosphorylated and translocates to the nucleus (*Brenna et al., 2019*), and (2) DARPP32 is phosphorylated and thus inhibits PKA activity (*Bibb et al., 1999*). Hence, before the light pulse at ZT14, the nucleus is supplied with PER2, which appears to be necessary for light-mediated behavioral phase delays (*Albrecht et al., 2001*; *Spoelstra et al., 2004*; *Chavan et al., 2016*). In parallel, the signaling pathway necessary to phosphorylate CREB is turned off. This state can then be dramatically changed when light is applied at ZT14 (*Figure 7b*) evoking glutamate and PACAP release at the synapses between the RHT and the SCN. Interaction between p35 and Cdk5 is abolished (*Figure 2*) thereby inactivating Cdk5 and stopping phosphorylation of PER2 and DARPP32. Since significant amounts of PER2 are already in the nucleus, this probably has no consequences on nuclear PER2 function. However, DARPP32 is not phosphorylated anymore and the block on PKA is released. At the same time, PKA becomes phosphorylated due to PACAP and cAMP signaling, leading to activation of Cav3.1 by PKA (*Figure 5f*; *Kim et al., 2006*). This results in CaMKII and CaMKIV phosphorylation and, ultimately, to the phosphorylation of CREB in the nucleus (*Figure 3*; *Matthews et al., 1994*). Phospho-CREB builds up a complex with CRTC1/CBP and PER2 (*Brenna et al., 2021*) to activate gene expression of the *Per1, Dec1, Sik1,* and *Gem* genes. In the activation complex the amount of PER2 present in the nucleus may at least in part affect the magnitude of the phase delay, which is depending on the time the light pulse is given. In conclusion, Cdk5 activity is gating both processes, the pre-light condition and the post-light condition, leading to a concerted

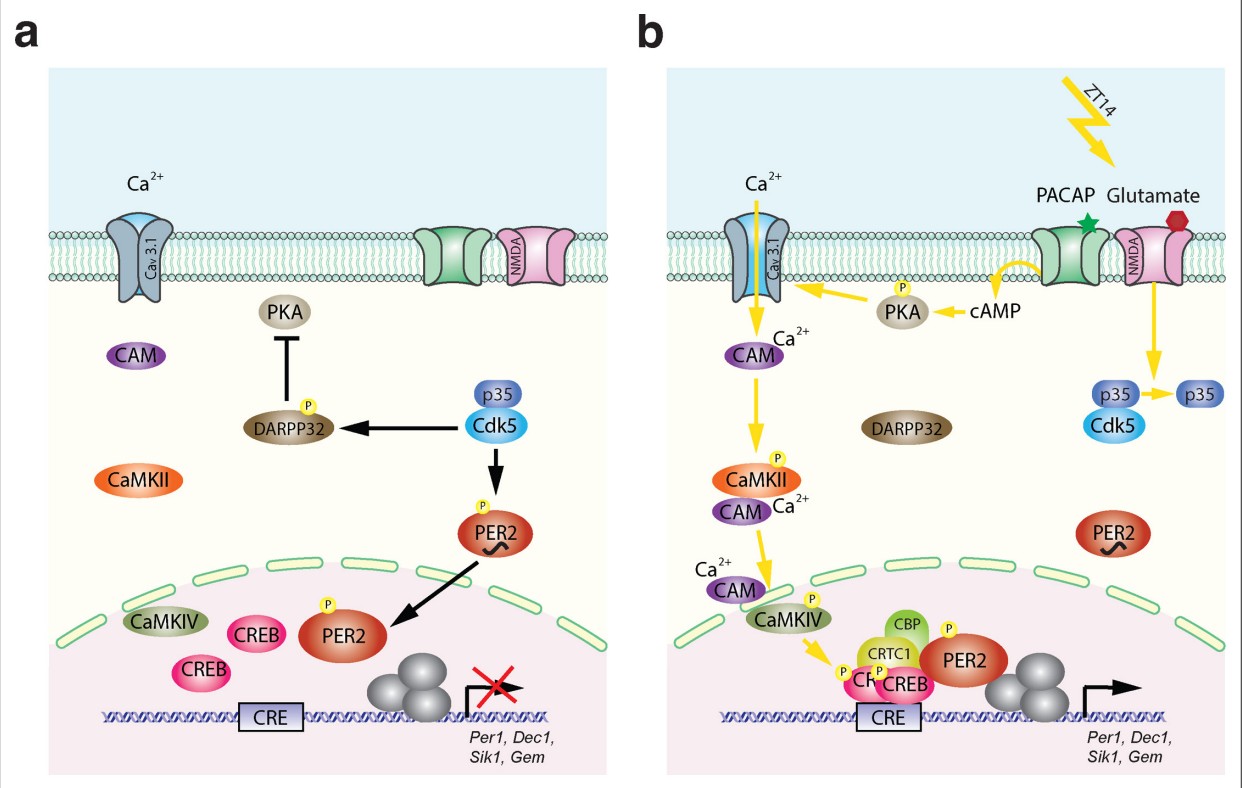

**Figure 7.** Model of Cdk5-gated light signal. (**a**) Cdk5 is active during the dark portion of the day. Active Cdk5 with its co-activator p35 phosphorylates PER2, which leads to stabilization and nuclear translocation of this protein that is abundant at zeitgeber time (ZT) 12. At the same time CDK5 phosphorylates DARPP32, which inhibits the PKA signaling pathway. (**b**) Light perceived in the dark phase at ZT14 leads to detachment of p35 from Cdk5 stopping Cdk5 activity. DARPP32 is not phosphorylated and hence can't inhibit PKA. PKA that is activated by the light signal is phosphorylated and can mediate cAMP response element-binding protein (CREB) phosphorylation via T-type calcium channels (Cav3.1) and the CaMK pathway leading to a transcriptionally active complex on the CRE element present in the promoters of many light-responsive genes such as *Per1*, *Dec1*, *Gem,* and *Sik1*. Overall, a light pulse at ZT14 will activate CREB phosphorylation and a protein complex will form. This complex needs phosphorylated PER2 that has accumulated in the nucleus between ZT12 and ZT14 to initiate the transcription of light-responsive genes. Both arms are necessary to build up a transcriptionally functional complex. Both arms depend on the presence and activity of CDK5, which therefore gates the light signal at ZT14. It is very likely that the amount of PER2 protein in the nucleus determines at least in part the magnitude of the phase delay, which depends on the timing of the light signal.

activation of a set of light-responsive genes that impinge on behavioral phase delays in response to nocturnal light exposure.

## Methods
### Animals and housing
All mice were housed with chow food (3432PX, Kliba-Nafag) and water ad libitum in transparent plastic cages (267 mm long, ×207 mm wide, ×140 mm high; Techniplast Makrolon type 2 1264C001) with a stainless steel wire lid (Techniplast 1264C116), kept in soundproof ventilated chambers at constant temperature (22 ± 2°C) and humidity (40–50%). All mice were entrained to a 12 hr LD cycle, and the time of day was expressed as ZT (ZT0 lights on, ZT12 lights off). Four-month-old 129/C57BL6 males were used for the experiments. Housing and experimental procedures were performed per the guidelines of the Schweizer Tierschutzgesetz, the declaration of Helsinki, and the ARRIVE guidelines. The state veterinarians of the Cantons of Fribourg and Bern approved the protocol (license numbers: 2021-19-FR; BE45/18; BE21/22).

## Locomotor activity monitoring

Locomotor activity parameters were analyzed by monitoring wheel-running activity, as described in *Riedel et al., 2018*, and calculated using the ClockLab software (Actimetrics). To analyze free-running rhythms, animals were entrained to LD 12:12 and released into constant darkness (DD). The internal period length was determined from a regression line drawn through the activity onsets of 10 days of stable rhythmicity under constant conditions, calculated using the respective inbuilt functions of the ClockLab software (Acquisition Version 3.208, Analysis Version 6.0.36). For better visualization of daily rhythms, locomotor activity records were double-plotted, which means that each day's activity is plotted twice, to the right and below that of the previous day. For the analysis of light-induced resetting, we used Aschoff type II and I protocols (*Jud et al., 2005*). For type II, mice maintained in LD 12:12 were subjected to a 15 min light pulse (LP, 500 lux) at ZT10 (no phase shift), 14 (phase delay), and 22 (phase advance). Subsequently, they were released into DD for 10 days, and phase shift was measured. For type I, mice maintained in DD were subjected to a 15 min light pulse at CT 10, 14, or 22. A circadian hour equals 1/24 of the endogenous period ($\tau$), calculated as follows: circadian hour = tau/24 hr. To calculate the CT hours corresponding the ZT, we followed these steps:

- $\text{CT12DayB} = \text{CT12 DayA} + \tau - 24\,\text{hr}$
- $\text{CTX0} - 12 = \text{CT12Day B} - X * 1\,\text{circadian hour}$ $[X = \text{CT12} - \text{CTx}]$
- $\text{CTX12} - 24 = \text{CT12Day B} + X * 1\,\text{circadian hour}$ $[X = \text{CTx} - \text{CT12}]$

The phase shift was determined by fitting a regression line through the activity onsets of at least 7 days under LD conditions before the light pulse and a second line through the activity onsets of at least 7 days under DD after the light pulse. The first 2 days after the administration of the light pulse were not considered for the calculation of the phase shift. The distance between the two regression lines determined the phase shift. Before starting any new protocol, mice were allowed to stabilize their circadian oscillator for 10 days. The corresponding figure legends indicate the number of animals used in the behavioral studies.

## Light pulse and tissue isolation

Light pulse (LP, 500 lux) was given at ZT14, and mice were sacrificed at appropriate indicated times. Brains were collected, and SCN tissue was isolated for western blot or RT-qPCR use. For immunofluorescence experiments, mice were perfused with 4% PFA and cryoprotected in 30% sucrose. Tissue isolation at ZT14 without a light pulse was used as light induction negative control.

## RNA extraction and cDNA synthesis

Total RNA was extracted from confluent 6 cm Petri dishes or frozen SCN tissue using the Microspin RNA II kit (Machery & Nagel, Düren, Germany) according to the manufacturer's instructions. 0.5 µg of total RNA was converted to single-strand cDNA in a total volume of 10 µL using the SuperScript IV VILO kit (Thermo Fisher Scientific, Waltham MA, USA) according to the manufacturer's instructions. The samples were diluted to 200 µL with pure water. 5 µL of each sample was mixed with 7.5 µL of KAPA probe fast universal real-time PCR master mix (Merck, Darmstadt, Germany) and 2.5 µL of the indicated primer/probe combinations. For the subsequent real-time PCR, a Rotorgene 6000 machine was used (QIAGEN, Hilden, Germany) and analyzed with the propriety software.

## qPCR primers
Per1

> FW: GGC ATG ATG CTG CTG ACC ACG
> RV: ACT GGG GCC ACC TCC AGT TC
> TM: FAM-TGG CCC TCC CTC ACC TTA GCC TGT TCC T-BHQ1

Per2

> FW: TCC ACA GCT ACA CCA CCC CTT A
> RV: TTT CTC CTC CAT GCA CTC CTG A
> TM: FAM-CCG CTG CAC ACA CTC CAG GGC G-BHQ1

## Dec1

FW: TGC AGA CAG GAG CGC ACA GT
RV: GCT TTGGGC AGG CAG GTA GGA
TM: FAM-TGG TTG CGC GCT GGG GAT CCG T-BHQ1

## Dec2

FW: ACA GAA TGG GGA GCG CTC TCT GAA
RV: TGA AAC CCC GAG TGG AAC GCA
TM: FAM-CGC CGG TCC AGG CCG ACT TGG A-BHQ1

## Bmal1

FW: GCA ATG CAA TGT CCA GGA AG
RV: GCT TCT GTG TAT GGG TTG GT
TM: FAM- ACC GTG CTA AGG ATG GCT GTT CAG CA-BHQ1

## eGFP

FW: CAT CTG CAC CAC CGG CAA GC
RV: GGT CGG GGT AGC GGC TGA A
TM: FAM- TGC CCG TGC CCT GGC CCA CC-BHQ1

## Fos

FW: GCC GGG GAC AGC CTT TCC TA
RV: TCT GCG CAA AAG TCC TGT GTG TTG A
TM: FAM-CCA GCC GAC TCC TTC TCC AGC ATG GGC-BHQ1

## Egr1

FW: CGG CAG CAG CGC CTT CAA T
RV: GGA CTC TGT GGT CAG GTG CTC AT
TM: FAM-CCT CAA GGG GAG CCG AGC GAA CAA CCC-BHQ1

## Sik1

FW: GGC TGC ACG ACC AGC AAT CG
RV: GGC GGT AGA AGA GTG GTG CTG TA
TM: FAM- TCC TGC ACC AGC AGA GGC TGC TCC AG-BHQ1

## Gem

FW: TGG GAA AAT AAG GGG GAG AA
RV: AGC TTG CAC GGT CTG TGA TA
TM: FAM- CCA CTG CAT GCA GGT CGG GGA TGC C-BHQ1

## Vip

FW: AGC AGA ACT TCA GCA CCC TAG ACA
RV: TCG GTG CCT CCT TGG CTG TT
TM: FAM- AGC CGG AAA GGC AGC CCT GCC T-BHQ1

*Tprkb* (normalization probe for *Tprkb*)

> FW: GGC TGG CAT CAG ACC CAC AGA
> RV: GGG CCC GTA GAG TCG GGA AA
> TM: FAM-CCT GCG TCT GCC CTC TGA GGG CTG-BHQ1

*Atp5h* (normalization probe for *Atp5h*)

> FW: TGC CCT GAA GAT TCC TGT GCC T
> RV: ACT CAG CAC AGC TCT TCA CAT CCT
> TM: FAM-TCT CCT CCT GGT CCA CCA GGG CTG TGT-BHQ1

*Sirt2* (normalization probe for *Sirt2*)

> FW: CAG GCC AGA CGG ACC CCT TC
> RV: AGG CCA CGT CCC TGT AAG CC
> TM: FAM- TGA TGG GCC TGG GAG GTG GCA TGG A-BHQ1

*Nono* (normalization probe for *Nono*)

> FW: TCT TTT CTC GGG ACG GTG GAG
> RV: GTC TGC CTC GCA GTC CTC ACT
> TM: FAM- CGT GCA GCG TCG CCC ATA CTC CGA GC-BHQ1

## Immunofluorescence

SCN cryosections (40 µM) were placed in a 24-well plate, washed three times with 1× TBS (0.1 M Tris/0.15 M NaCl) and 2× SSC (0.3 M NaCl/0.03 M tri-Na-citrate pH 7). Antigen retrieval was performed with 2× SSC heating to 85°C for 30 min. Then, sections were washed twice in 2× SSC and three times in 1× TBS pH 7.5 before blocking them for 1.5 hr in 10% fetal bovine serum (Gibco)/0.1% Triton X-100/1× TBS at room temperature (RT). If the recipient species for some raised antibody was the mouse, we performed a Mouse on Mouse (MOM; Ab269452) blocking (2 hr) before 10% FBS to block endogenous mouse immunoglobulins in a mouse tissue section. After the blocking, the primary antibodies (*Table 1*), diluted in 1% FBS/0.1% Triton X-100/1× TBS, were added to the sections and incubated overnight at 4°C. The next day, sections were washed with 1× TBS and incubated with the appropriate fluorescent secondary antibodies diluted 1:500 in 1% FBS/0.1% Triton X-100/1× TBS for

**Table 1.** Antibodies used for the immunostainings.

| Antibody | Species | Company | Catalog number | Dilution |
|---|---|---|---|---|
| Anti-Cdk5 clone 2H6 | Mouse | Origene | CF500397 | 1:100 |
| Anti-GFP | Rabbit | Abcam | ab6556 | 1:500 |
| Anti-Creb | Rabbit | Cell Signaling | D76D11 | 1:200 |
| Anti-Creb (pSer133) | Rabbit | Abcam | Ab32096 | 1:500 |
| Anti-CaMKII | Rabbit | Abcam | Ab52470 | 1:200 |
| Anti-CaMKII (pThr286) | Mouse | Invitrogen | MA1-047 | 1:100 |
| Anti-CaMKIV | Rabbit | Abcam | Ab3557 | 1:200 |
| Anti-CaMKIV (pThr196/200) | Rabbit | Invitrogen | PA5-105011 | 1:100 |
| Anti-CaV3.1 | Rabbit | Invitrogen | PA5-50635 | 1:100 |
| Anti-Calmodulin | Mouse | Invitrogen | MA3-917 | 1:100 |
| Anti-PKA (pT197) | Rabbit | Abcam | Ab75991 | 1:100 |
| Anti-Darpp32 (pThr75) | Rabbit | Invitrogen | PA5-105037 | 1:100 |

3 hr at RT (Alexa Fluor 488-AffiniPure Donkey Anti-Rabbit IgG (H+L) no. 711-545-152, Lot: 132876, Alexa Fluor 647-AffiniPure Donkey Anti-Mouse IgG (H+L) no. 715-605-150, Lot: 131725, Alexa Fluor 647-AffiniPure Donkey Anti-Rabbit IgG (H+L) no. 711-602-152, Lot: 136317 and all from Jackson ImmunoResearch). Tissue sections were stained with DAPI (1:5000 in PBS; Roche) for 15 min. Finally, the tissue sections were rewashed twice in 1× TBS and mounted on glass microscope slides. Fluorescent images were taken using a confocal microscope (Leica TCS SP5), and pictures were taken with a magnification of ×63 with or without indicated additional zoom. Images were processed with the Leica Application Suite Advanced Fluorescence 2.7.3.9723. Immunostained sections were quantified using ImageJ version 1.49. Statistical analysis was performed on three animals per treatment.

## AAV production and stereotaxic injections

Experiments were performed as previously described (*Brenna et al., 2019*). Stereotaxic injections were performed on 4- to 5-month-old mice under isoflurane anesthesia using a stereotaxic apparatus (Stoelting). The brain was exposed by craniotomy, and the Bregma was used as a reference point for all coordinates. AAVs were injected bilaterally into the SCN (Bregma: anterior-posterior [AP] − 0.40 mm; medial-lateral [ML] ±0.00 mm; dorsal-ventral (DV) − 5.7 mm, angle ±3°) using a hydraulic manipulator (Narishige: MO-10 one-axis oil hydraulic micromanipulator, http://products.narishige-group.com/group1/MO-10/electro/english.html) at a rate of 40 nL/min through a pulled glass pipette (Drummond, 10 µL glass micropipette; Cat. number: 5-000-1001-X10). The pipette was first raised 0.1 mm to allow the spread of the AAVs and later withdrawn 5 min after the end of the injection. After surgery, mice were allowed to recover for 2 weeks and entrained to LD 12:12 before behavior and molecular investigations. Based on health after surgery, sick animals were excluded. Virus injected animals that did not show shorter period were excluded (injection probably not correct).

The injected viruses were:

- SsAAV-9/2-hSyn1-chI[mouse(sh*Cdk5*)]-EGFP-WPRE-SV40p(A)
- ssAAV-9/2-hSyn1-chI[1x(shNS)]-EGFP-WPRE-SV40p(A)

## Protein extraction from SCN tissue

The protocol was a modified version of what was published before (*Brenna et al., 2021*). Isolated SCNs obtained from four different mice were pooled according to the indicated condition (either dark or 15 min after the light pulse). The pooled tissues were frozen in liquid N2 and resuspended in a brain-specific lysis buffer (50 mM Tris-HCl pH 7.4, 150 mM NaCl, 0.25% SDS, 0.25% sodium deoxycholate, 1 mM EDTA). They were homogenized using a pellet pestle, kept on ice for 30 min and vortexed for 30 s, followed by N2 freezing. Frozen samples were left to melt on ice. The samples were sonicated (10 s, 15% amplitude) and centrifuged for 20 min at 16,000×*g* at 4°C. The supernatant was collected in new tubes, and the pellet was discarded.

## Immunoprecipitation

The protocol was described before (*Brenna et al., 2019*). The protein extract was diluted with the appropriate lysis buffer in a final volume of 250 µL and immunoprecipitated using the indicated antibody (ratio 1:50). The reaction was kept at 4°C overnight on a rotary shaker. The day after, samples were captured with 50 µL of 50% (wt/vol) protein-A agarose beads (Roche), and the reaction was kept at 4°C for 3 hr on a rotary shaker. Before use, beads were washed thrice with the appropriate protein

**Table 2.** Antibodies used for the western blots.

| Antibody | Species | Company | Catalog number | Dilution |
|---|---|---|---|---|
| Anti-Cdk5 | Rabbit | Cell Signaling | D1F7M | 1:3000 |
| Anti-Creb | Rabbit | Cell Signaling | D76D11 | 1:3000 |
| Anti-Creb (pSer133) | Rabbit | Abcam | Ab32096 | 1:1000 |
| Anti-PKA (pT197) | Rabbit | Abcam | Ab75991 | 1:1000 |
| Anti-p35 | Rabbit | Invitrogen | MA5-14834 | 1:1000 |
| Anti-CaMKIV (pThr196/200) | Rabbit | Invitrogen | PA5-105011 | 1:1000 |

buffer and resuspended in the same buffer (50% wt/vol). The beads were collected by centrifugation and washed three times with NP-40 buffer (100 mM Tris-HCl pH 7.5, 150 mM NaCl, 2 mM EDTA, 0.1% NP-40). After the final wash, beads were resuspended in 2% SDS 10%, glycerol, 63 mM Tris-HCl pH 6.8, and proteins were eluted for 15 min at RT. Laemmli buffer was finally added, and samples were boiled for 5 min at 95°C and loaded onto 10% SDS-PAGE gels.

## Western blot

Circa 40 μg of protein was loaded onto 10% SDS-PAGE gel and run at 100 V for 2 hr. Protein migration was followed by a semidry transfer (40 mA, 1 hr 30 s) using Hybond ECL nitrocellulose. We subsequently performed red Ponceau staining (0.1% of Ponceau S dye and 5%) on the membrane to confirm the successful transfer. The list of antibodies used in the paper is shown in *Table 2*. The membrane was washed with TBS 1×/Tween 0.1% and blocked with TBS 1×/BSA 5%/Tween 0.1% for 1 hr. After washing, the membrane was blotted with the appropriate primary antibodies overnight. The day after, membranes were washed three times with TBS 1×/Tween 0.1%, followed by secondary antibody immunoblotting for 1 hr at room temperature. The densitometric signal was digitally acquired with the Azure Biosystem.

## In vitro kinase assay using immunoprecipitated Cdk5 from SCN

The protocol is the same as before (*Brenna et al., 2019*). CDK5 was immunoprecipitated (4°C overnight with 2× Protein A agarose [Sigma-Aldrich]) from SCN samples at ZT14 in the dark or after a light pulse (LP, circa 500 lux) of 15 min. After immunoprecipitation, samples were diluted in a washing buffer and split into two halves. One-half of the IP was used for an in vitro kinase assay. Briefly, 1 μg of histone H1 (Sigma-Aldrich) was added to the immunoprecipitated CDK5. Assays were carried out in reaction buffer (30 mM HEPES, pH 7.2, 10 mM $MgCl_2$, and 1 mM DTT) containing [γ-$^{32}$P] ATP (10 Ci) at 30°C for 1 hr and then terminated by adding SDS sample buffer and boiling for 5 min. Samples were subjected to 15% SDS-PAGE, stained by Coomassie Brilliant Blue, and dried, and then phosphorylated histone H1 was detected by autoradiography. The other half of the IP was used for western blotting to determine the total CDK5 immunoprecipitated from the SCN samples. The following formula was used to quantify the kinase activity at each time point: ([$^{32}$P] H1/total H1)/CDK5 $_{IP protein}$.

## Cell culture

NIH3T3 wt and CRISPR/Cas9 *Cdk5* ko (*Brenna et al., 2019*) mouse fibroblast cells (ATCCRCRL-1658) were maintained in Dulbecco's modified Eagle's medium (DMEM), containing 10% fetal calf serum and 100 U/mL penicillin-streptomycin at 37°C in a humidified atmosphere containing 5% $CO_2$. For any experiment, cells were synchronized with forskolin (100 μM).

## Surgical procedure for fiber photometry recordings

Animals previously infected either with AAV9-hSyn1-chl[1x(shNS)]-jGCaMP7b-WPRE-SV40p(A) (*scramble*) or AAV9-hSyn1-chl[mouse(sh*Cdk5*)]-jGCaMP7b -WPRE-SV40p(A) (*shCDK5*) were injected with Metacam (Meloxicam, 5 mg/kg s.c.) for analgesia, then anesthetized with 1.5–2% isoflurane/$O_2$ mix. Mice were placed in a Kopf digital stereotactic frame. Their body temperature was kept constant at 37°C via a feedback-coupled heating device (Panlab/Harvard Apparatus), and their eyes were covered with ointment (Bepanthen Augen- und Nasensalbe, Bayer). After the skin incision (formerly prepared aseptically), the skull bone was cleaned with saline to remove the remaining tissue. A small craniotomy to target the SCN was drilled into the skull (Micro-Drill from Harvard Apparatus with burrs of 0.7 mm diameter from Fine Science Tools), and the dura was carefully removed. An optical fiber implant (400 μm, 0.5 NA Core Multimode Optical Fiber, FT400ERT, inserted into ceramic ferrules, 2.5 mm OD; 440 μm ID, Thorlabs) was slowly implanted above the SCN to allow for imaging of GCaMP7b signals (AP: +0.40; ML: ±0.0; DV: –5.3; angle ±4°). One stainless steel screw was inserted into the skull over the cerebellum for stability purposes. The implant was then secured to the skull with dental cement (Paladur, Kulzer). After surgical procedures, mice were allowed to recover for 1 week and finally tethered with an optical patch cord.

## Fiber photometry experimental design

GCaMP7b was excited with a blue LED (Doric, LED driver, assembled with 470 nm) at 480 Hz, and emission was sampled at 2000 Hz with a photodetector (Doric, DFD_FPA_FC) through a fluorescence MiniCube (Doric, ilFMC6_IE(400-410)_E1(460–490)_F1(500–540)_E2(555–570)_F2(580–680)_S) and digitized with a national instruments USB-6002 DAQ device. Fiber photometry recordings were acquired using custom-written scripts in LabVIEW on a computer. All the recordings were started about 15 min before ZT14 to stabilize the fluorescent signal. For every trial at ZT14 a constant light pulse of 10,000 lux (Daylight Lamp) was manually turned on for 15 min (±20 s), and the recording was stopped 15 min after the light was switched off. To allow mice to restore their CT to the 12 hr LD cycle, the intertrial interval was at least 10 days. A patch cord was connected to the light source and the photometry system to align the light pulse to the photometry recording.

## In vivo calcium imaging, data processing, and analysis

The data were subdivided into control (*scramble*) and experimental (*shCDK5*) groups.

The fluorescent signal was demodulated in the frequency band of 470–490 Hz at 10 Hz acquisition rate. Due to GCaMP7b variable photobleaching (i.e. the loss of fluorescence intensity as a function of light exposure), we filtered the demodulated signal using a three order Savitzky-Golay filter (every 100 s), and detrended it using a hug-line. We then calculated the $\Delta F/F_0$ as follows:

$$\frac{\Delta F}{F_0} = \frac{detrended\ signal - 1^{st}percentile\ (detrended\ signal)}{99^{th}percentile\ (detrended\ signal) - 1^{st}percentile\ (detrended\ signal)}$$

Finally, the $\Delta F/F_0$ was cut to the light pulse as follows: (a) 5 min before the light pulse, (b) 15 min during the light pulse, and (c) 15 min after the light. To exclude differences in the duration of the light pulse (±20 s), the period analyzed was of 14 min. All data processing was performed using custom-written MATLAB scripts.

## Live FRET imaging

The protocol was performed as before (*Brenna et al., 2021*). The following plasmid was used for the project: ICAP-NLS Vector carrying (*Brenna et al., 2021*). Transfected NIH3T3 cells were starved for 4 hr with 0.5% FBS DMEM. Subsequently, cells were washed twice with 1× HBTS without $CaCl_2$ and $MgCl_2$ (25 mM HEPES, 119 mM NaCl, 6 g/L glucose, 5 mM KCl) and resuspended in the same buffer. NIH3T3 cells were imaged using an inverted epifluorescence microscope (Leica DMI6000B) with an HCX PL Fluotar 5×/0.15 CORR dry objective, a Leica DFC360FX CCD camera (1.4 M pixels, 20 fps), and EL6000 Light Source, and equipped with fast filter wheels for FRET imaging. Excitation filters for CFP and FRET: 427 nm (BP 427/10). Emission filters for CFP: 472 (BP 472/30) and FRET: 542 nm (BP 542/27). Dichroic mirror: RCY 440/520. One frame every 20 s was acquired for at least 90 cycles (0.05 Hz frequency), and the recording lasted at least 30 min. The baseline response in the presence of HBTS was recorded for 2 min and 40 s. At min 3:00, 100 µM forskolin, 2 mM $CaCl_2$, and 2 mM $MgCl_2$ were added to the cells. The time-lapse recordings were analyzed using LAS X software (Leica). Two regions of interest (ROI) were randomly selected for each cell, and 50 cells per plate were chosen randomly. A first ROI delimiting the background and a second ROI including the cell nucleus of NIH3T3 cell expressing NLS KIDKIX were used per cell. The ROI background values were subtracted from the ROI cell values for each channel. For baseline normalization, the FRET ratio R was expressed as a ΔR/R, where ΔR is R–R0, and R0 is the average of R over the last 120 s prior stimulus.

## Statistical analysis

Sample size determination was done using public G*Power software 3.1.9.7.

Statistical analysis of all experiments was performed using GraphPad Prism6 software. Depending on the data type, either an unpaired t-test or one- or two-way ANOVA with Bonferroni or Tukey's post hoc test was performed. Values considered significantly different are highlighted [$p<0.05$ (*), $p<0.01$ (**), or $p<0.001$ (***)].

Data were compared via two-way repeated-measures ANOVA with post hoc Bonferroni's corrections for multiple comparisons. Data distribution was assumed to be normal, but this was not formally tested. All data are displayed as mean ± standard error of the mean (SEM). No power calculations

were performed to determine sample sizes, but similarly sized cohorts were used in previous studies. The experimenters were not blind to the conditions when acquiring or analyzing the data.

Sample numbers are indicated in the corresponding figure legends, and test details are only reported for significant results. Figures were prepared in Adobe Illustrator 2022.

## Acknowledgements

Technical assistance by Antoinette Hayoz and Maude Marmy is acknowledged. This work was supported by the Swiss National Science Foundation (SNF) 310030_219880/1 to UA, 310030_197607 to DAG, 310030_219438/1 to ZY, the Inselspital University Hospital Bern, the European Research Council CoG-725850 to AA and the States of Berne and Fribourg.

## Additional information

### Funding

| Funder | Grant reference number | Author |
| --- | --- | --- |
| Schweizerischer Nationalfonds zur Förderung der Wissenschaftlichen Forschung | 310030_219438/1 | Antoine Adamantidis |
| Schweizerischer Nationalfonds zur Förderung der Wissenschaftlichen Forschung | 310030_197607 | Dominique A Glauser |
| European Research Council | CoG-725850 | Antoine Adamantidis |
| Schweizerischer Nationalfonds zur Förderung der Wissenschaftlichen Forschung | 310030_219880/1 | Urs Albrecht |
| University of Fribourg | 310030_219438/1 | Zhihong Yang |

The funders had no role in study design, data collection and interpretation, or the decision to submit the work for publication.

### Author contributions

Andrea Brenna, Conceptualization, Data curation, Formal analysis, Validation, Investigation, Visualization, Methodology, Writing – original draft; Micaela Borsa, Data curation, Formal analysis, Validation, Investigation, Visualization, Methodology, Writing – review and editing; Gabriella Saro, Data curation, Formal analysis, Validation, Investigation, Writing – review and editing; Jürgen A Ripperger, Validation, Investigation, Methodology, Writing – review and editing; Dominique A Glauser, Resources, Validation, Methodology, Writing – review and editing; Zhihong Yang, Resources, Funding acquisition, Writing – review and editing; Antoine Adamantidis, Resources, Supervision, Funding acquisition, Validation, Methodology, Writing – review and editing; Urs Albrecht, Conceptualization, Resources, Supervision, Funding acquisition, Validation, Visualization, Writing – original draft, Project administration, Writing – review and editing

### Author ORCIDs

Andrea Brenna ⓘ https://orcid.org/0000-0002-8542-9855
Micaela Borsa ⓘ https://orcid.org/0000-0001-7634-7738
Jürgen A Ripperger ⓘ https://orcid.org/0000-0002-9345-5172
Dominique A Glauser ⓘ https://orcid.org/0000-0002-3228-7304
Zhihong Yang ⓘ https://orcid.org/0000-0002-4133-5099
Urs Albrecht ⓘ https://orcid.org/0000-0002-0663-8676

## Ethics

Housing and experimental procedures were performed per the guidelines of the Schweizer Tierschutzgesetz, the declaration of Helsinki and the ARRIVE guidelines. The state veterinarians of the Cantons of Fribourg and Bern approved the protocol (license numbers: 2021-19-FR; BE45/18; BE21/22).

Reviewer #1 (Public review): https://doi.org/10.7554/eLife.97029.3.sa1

Reviewer #2 (Public review): https://doi.org/10.7554/eLife.97029.3.sa2

Author response https://doi.org/10.7554/eLife.97029.3.sa3

# Additional files

## Supplementary files

MDAR checklist

## Data availability

All data generated or analysed during this study are deposited in Dryad. Non-commercial biological materials are provided upon request to the corresponding author.

The following dataset was generated:

| Author(s) | Year | Dataset title | Dataset URL | Database and Identifier |
|---|---|---|---|---|
| Brenna A, Borsa M, Saro G, Ripperger J, Glauser D, Yang Z, Adamantidis A, Albrecht U | 2025 | Data from: Cyclin-dependent kinase 5 (Cdk5) activity is modulated by light and gates rapid phase shifts of the circadian clock | https://doi.org/10.5061/dryad.8sf7m0d0d | Dryad Digital Repository, 10.5061/dryad.8sf7m0d0d |

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
