## [Editor Report · eLife Assessment]

This **important** chronobiological study in mice suggests that light modulated activity of Cdk5 activity on the PKA-CaMK-CREB signaling pathway provides missing molecular mechanistic details to understand light-induced circadian clock phase delays during the early night, but not for phase advances in the morning. The authors provide **convincing** evidence bridging from behavioral to molecular/cellular experiments to neural activity imaging.

---

## [Referee Report · Reviewer #1 (Public review)]

In the manuscript Cyclin-dependent kinase 5 (Cdk5) activity is modulated by light and gates rapid phase shifts of the circadian clock Brenna et al., study the role of Cdk5 on circadian rhythms, the authors aim to elucidate the role of Cyclin-Dependent Kinase 5 (Cdk5) in modulating circadian rhythms, particularly in response to light cues. They hypothesized that Cdk5 acts as a gatekeeper, regulating the sensitivity of the circadian clock to light-induced phase shifts.

Strengths:

• Novelty: The study presents a novel mechanism by which Cdk5 influences circadian rhythms, particularly its role in modulating the light-induced phase-shifting response.

• Experiments: The authors have employed a combination of molecular, cellular, and behavioural techniques, including genetic manipulations, biochemical assays, and electrophysiology, to investigate the role of Cdk5. The set of experiments performed in this work is non-trivial, done to a high standard and the additional experiments, data and textual alterations presented following the 1st round of review needs to be lauded.

• Data: The data is well-presented in clear figures and appropriately described in the text.

Weaknesses:

• Although I found the data on Cdk5 gating light responses highly convincing there could be additional mechanisms which the authors have duly acknowledged and discussed in their text.

In my assessment, the authors have convincingly demonstrated that Cdk5 plays a critical role in gating the light-induced phase-shifting response of the circadian clock. Their results strongly support their conclusions, as evidenced by their findings:

This study provides valuable insights into the molecular mechanisms underlying circadian rhythm regulation and the impact of light on the circadian clock. The findings have the potential to influence future research in the field of chronobiology and may have implications for understanding and treating circadian rhythm disorders.

The methods and data presented in this study are valuable to the field and can be used to further investigate the role of Cdk5 and other signalling pathways in circadian rhythm regulation.

Broader context

The circadian clock is a fundamental biological process that regulates various physiological functions, including sleep-wake cycles, hormone secretion, and metabolism. Disruptions to the circadian clock have been linked to a variety of health problems, such as sleep disorders, metabolic disorders, and cancer. Understanding the molecular mechanisms that underlie circadian rhythm regulation is essential for developing effective treatments for these disorders.

All in all, I have no reservations regarding the manuscript titled "Cyclin-dependent kinase 5 (Cdk5) activity is modulated by light and gates rapid phase shifts of the circadian clock by Brenna et al. After consideration of the authors' revisions, I believe the manuscript has been significantly improved. I commend the authors for their diligence in addressing the reviewers' comments and for the quality of their research.

---

## [Referee Report · Reviewer #2 (Public review)]

Summary:

Definition of the role of CdK5 in circadian locator activity and light induced neural activity in the mouse SCN in-vivo revealing its mode of action through PKA-CaMK-CREB signaling pathway.

Strengths:

The experimental approaches are carried from in-vivo, to cellular and molecular level and provide first evidence for the specific involvement of CdK5 in light-induced phase advance of the free-running rhythm.

Weaknesses:

The behavioral analyses are limited to some selected parameters.

Downstream effects on circadian oscillation of gene expression and physiological functions in other brain regions, organs is missing.

Comments on revisions:

I am happy with the manuscript in its present form.

---

## [Author Response]

The following is the authors’ response to the original reviews.

**Reply to reviewer comments:**

(1) Given the interpretations of this study hinge on the specificity of the antibodies used in immune fluorescence, the authors should provide full western-blot images of all their antibodies in supplementary information.

The commercial antibodies have been validated by the provider.

Additionally, we did our own tests. Of note is that proper validation of any antibody is only possible by using a knockout mouse for each protein analyzed (i.e. for pPKA wt vs. *pka* ko mice). This is not possible, because we do not have all these knock-out strains. However, specific proteins like pPKA, pCAMKII, and pCAMKIV are known to be increased by a light pulse. We show by western blot that pPKA (Fig. 2a, b) and pCamKII (Fig. S2a, b) are increased in wt animals mirroring what we observed in the immunofluorescence. These results suggest that the signal is specific to these antibodies. We provide a full panel of western blots, including the other proteins studied by immunofluorescence such as pCamKIV, pCREB, CaV 3.1, and pDARP32 and show that they detect a protein of the expected size. Full Western-blots mentioned in the manuscript are shown in Supplementary Figure 7. Below are additional validations of antibodies used in the immunofluorescence experiments.

**Author response image 2. sa3fig2:** 

(2) The explanation in the results section surrounding Fig. 4 seems to be specific for the representative trace rather than the group. Specifically, does the following statement apply to all the replicates? " A Ca2+ transient was observed right before the light was given at ZT14 (Fig. 4b), which showed the same magnitude as those observed during and after the light stimulus".If not this should be corrected.

We have replaced now Fig. 4b with an average trace of all experiments. The individual traces can be seen in supplementary figure 4d.

(3) Are lines 236 -244 and figure 5A/B demonstrating shCDK5 being similar to no-calcium or EGTA conditions at the level of CREB not contradicting Figure 3 which argues that the reason behind the increase in CAMK-phosphorylation and pCREB following shCDK5 is increased basal calcium? If this is the case then why does removing the external calcium phenocopy shCDK5 in these cells? The authors need to clarify this and give an explanation.(4) The authors should explain why they see an equivalent level (or more) of CREB activation, 5 minutes following forskolin activation in Ca2+-free condition (apparent in the case of shCDK5 and EGTA) in the FRET assay. Does this not imply PKA is the most likely candidate mediating this reaction at this stage? Given this interaction has been demonstrated in multiple (other) experiments including in vitro isolated enzyme experiments involving CREB and PKA (E.G. fig 6A in PMID: 2900470) an absence of p-PKA pulldown is not sufficient to justify the non-involvement of PKA (PMID: 22583753). This statement needs support in the form of positive data or acknowledging the limitations in the text (conditions, single technique, etc).(5) The authors should better explain the fret pairs used in the experiments involving ICAP for the reader's benefit - a reduction in fluorescence as a function of CREB activation is non-intuitive.

We answer all three questions (3-5) together since they belong to the same concept.

(1) How FRET works.

The Forster resonance energy transfer (FRET) technique is widely used to investigate molecular interactions between proteins such as CREB: CBP in living cells. We used a sensor called ICAP (an Indicator of CREB Activation due to Phosphorylation) published by Friedrich and colleagues in 2010

(https://doi.org/10.1074/jbc.M110.124545). The sensor is composed of three different elements: (1) the KID domain of CREB containing the Ser-133, which is phosphorylated upon forskolin induction in our experimental setup, (2) the KIX domain of CBP, which is responsible for the dimerization with phospho-CREB and (3) a short linker that separates the KID with the KIX domain. KID is flanked by a cyan fluorescent protein (CFP), while KIX is flanked by a yellow fluorescent protein (YFP). When KID is not phosphorylated, the ICAP conformation allows CFP - stimulated by blue UV light - to transfer energy to YFP, producing FRET resulting in yellow light emission. Therefore, the ratiometric analysis FRET/CFP shows FRET > CFP. After a stimulus (forskolin), the serine-133 in KID is phosphorylated and KID can bind to KIX. The dimerization separates CFP from YFP, resulting in decreased FRET and increased CFP-dependent blue light emission (see Author response image 3 below). Therefore, the ratiometric analysis FRET/CFP shows FRET<CFP over time (usually within 20’ after the forskolin stimulus).

**Author response image 3. sa3fig3:** FRET model. On the left is a schematic representation of how ICAP works. On the right, an example of the quantified FRET decrease associated with increased KID: KIX interaction.

(2) The ‘apparent’ contradiction between Figure 5A and Fig 3.

As mentioned before, the chosen FRET method is ratiometric, meaning that a relative FRET signal in fluorescence is measured compared to the baseline (absence of forskolin, assay buffer). The FRET experiment can only tell whether there is a change in the phosphorylation state of KID during the live imaging comparing the baseline to the period after the forskolin treatment. The result produces a delta [(time after forskolin)(baseline)]. The higher the delta, the more KID is phosphorylated after forskolin treatment. If KID phosphorylation is not increased compared to the baseline, the FRET signal tends to return to the baseline with a reduced delta [(time after forskolin)-(baseline)]. Therefore, the experiment does not tell at the quantitative level the amount of KID (CREB domain) phosphorylation before the stimulus. It only tells whether after the stimulus the phosphorylation is increased producing or not a delta. This means that the lack of delta can be caused by: (A) high KID phosphorylation in the baseline which does not further increase after the forskolin stimulus; (B) very low KID phosphorylation in the baseline which does not increase after the forskolin stimulus. In Fig. 5A, wt cells (orange trace, lines, and double arrow) show a higher delta compared to the ko cells (blue trace, lines, and double arrow). The result indicated that the phosphorylation of CREB (KID domain) is increased after the forskolin stimulus only in the wt. To that extent, the results are in line with the experiment that we show in Figure 3. Indeed, the increased delta in CREB phosphorylation is observed only in the scramble animals, where it is lost in the ko (the blue double arrow indicates the delta in the scramble).

**Author response image 4. sa3fig4:** 

(3) The FRET signal within 3 minutes after forskolin stimulation

The signal mentioned by the reviewers at 5’ is an artifact given by the light diffraction promoted by the addition of Forskolin in DMSO which propagates through the plate. The same effect is observed in the only DMSO treatment (Fig.S5). Therefore, it needs not to be taken into account. The amplitude of this signal in this window of time is due to many independent variables (buffer composition, cell shape, room temperature, pipetting), therefore it is not possible to speculate any consideration about it. We never consider this time window for describing our results.

**Author response image 5. sa3fig5:** 

(4) Role of PKA and considerations about experiments performed in Fig. 5a and b

To answer the question about the role of PKA, we believe it is a pivotal player. Our results indicate that PKA might promote CaV3.1, the entrance of calcium, and therefore, CAM Kinase pathway activation leading to CREB phosphorylation (Fig. 5). However, if the calcium is depleted, even a channel activation mediated by PKA cannot propagate the signal. For that reason, when we deplete calcium in wt cells as we do in the experiment performed in Figure 5B the activation of PKA alone cannot promote the CREB phosphorylation associated with a reduction of the FRET signal. As mentioned before, the FRET method gives a binary answer. It means either a higher or lower delta comparing time after forskolin to baseline. It cannot give stoichiometric info about the level of calcium and/or phosphorylation in the baseline. To that extent, the FRET experiment in Figure 5A cannot be connected to the experiment in Figure 5B. The method is the same, but the scientific questions are different. In Figure 5A we demonstrate that CDK5 plays a role in the PKA activation pathway. In Figure 5B we demonstrate that the general pathway needs calcium.

We modified the text accordingly.

(6) The presentation of the data in Figure 6 seems to be divergent from the rest of the data presentations. Please make it more consistent and also provide more explanations. Specifically, the authors suggest increased P-CREB nuclear localization (and an increase in phosphorylated PKA/CAMK) following shCDK5. Won't this lead to an increase in Per1, Dec1, cFos, and Sik1 basally (pre-light pulse)?

We followed the reviewer's suggestion and present data in Figure 6 as done before in the manuscript. The reviewers should also consider our papers published before (Brenna et al., 2019; Brenna et al., 2021). In these papers, we demonstrate two important concepts that are in line with this manuscript. First, the lack of CDK5 promotes PER2 degradation and lack of nuclear translocation (Brenna et al., 2019). Second, PER2 plays a scaffold role in promoting the formation of the CREB transcriptional complex involved in the regulation of the expression of light-dependent genes (Brenna et al., 2021). Therefore, the take-home message here is that even if a lack of Cdk5 promotes a higher basal level of CREB phosphorylation, it also promotes PER2 degradation. Therefore, without PER2, the CREB-dependent gene expression is reduced. For this reason, we say that CDK5 gates phase shift (via PKA-CAM Kinases-CREB axis) of the circadian clock (via PER2).

(7) The authors should discuss why calcium-sensitive phosphatases such as PP2A (PMID: 23752926) or calcineurin (PMID: 10217279) are not considered candidates for dephosphorylation of DARPP32 as these are described previously (CDK5) and conditions of increased calcium as seen here would favour these enzymes. The phospho-T75 data are supportive, but such additional discussion could be important given the past demonstrations.

We thank the reviewers for the great insight. The pathway that promotes the T75 phosphorylation/dephosphorylation indeed includes many players as calcineurin and PPA2A. We mention this in the discussion now as follows:

However, phosphatases such as PP2A and calcineurin, which de-phosphorylate DARPP32 including the Cdk5 phosphorylation site, may be involved in this process as well (Girault and Nairn, 2021). Upon light treatment and increase of Ca2+ these phosphatases would dephosphorylate DARPP32 and thereby inactivate it, leading to PKA activation. This process may occur in parallel to the Cdk5 regulation of DARPP32 contributing to a sustained activation of the light signaling pathway via PKA activation.

(8) additional details on the knock-downs would be helpful:- the relative amount of reduction in gene expression upon shRNA treatment should be provided - How was the exact viral delivery and reduction in shRNA-induced knock-down confirmed for the individual animals?

The validation of Cdk5 knockdown was widely performed in the previous paper (Brenna et al., 2019, Fig2-Fig supp1, and Fig3-Fig suppl2). We used the same mice. We confirmed the goodness of the silencing also in the supp figure 1A of the current paper.

(9) The authors only focus on male mice. This is rather incomplete, as it leaves away an important half of biological reality. Testing relevant aspects of the work in female mice would close this significant gap and also increase the number of biological replicates, which can still be considered relatively low.

We thank the reviewers for the suggestion. We injected female mice and performed the Ashoff type-II light pulse experiment at ZT14 and observe the same phenotype as for male mice. This is stated now in the paper and the data are shown in supplemental figure 1 e-f.

(10) Given the roles of CdK5 in circadian clock period length regulation, but also light-induced phase delays, it would be interesting for a broader audience to discuss possible expectations of CdK5's roles, e.g.(a) How will other circadian parameters, eg. activity bouts (numbers, length, activity onset/ offset) be affected?(b) How does that relate to sleep, sleep phases?(c) What is the expected impact on other physiological rhythms, eg food intake, cortisol levels?(d) What are the expected effects on circadian oscillation of gene expression in other brain regions, organs?

We thank the reviewers for the observations.

(a) The activity was discussed in the previous paper (Brenna et al. 2019). ShCdk5 mice show a reduced activity in both DD and LD 12:12 compared to wt, mirroring the Per2 brdm phenotype (Figure- Suppl3, with the difference mostly observed at night time Figure 2-suppl4).

We also demonstrate in Figure 1 - figure supplement 1b, c of the current paper that light pulse does not affect the period length either in scramble mice or in sh Cdk5.

(b) We performed preliminary experiments with SCN shCdk5 knock-down animals and compared them to scr control mice using the Piezo sleep system. Total sleep was not different, however during the dark phase shCdk5 animals tended to sleep a bit more, similar to the neuronal Per2 KO animals (Wendrich et al., 2023 https://doi.org/10.3390/clockssleep5020017). After sleep-deprivation no differences were observed between shCdk5 and scr animals. This was comparable to the neuronal Per2 KO animals that also showed no phenotype after sleep deprivation.

(c) and (d) We did not investigate food intake, cortisol, or other parameters involving peripheral clocks. We did not investigate the gene expression in other brain regions because the SCN is the main brain region involved in the regulation of the circadian clock phase shift. However future studies will address these questions.